# Impaired memory B-cell recall responses in the elderly following recurrent influenza vaccination

**Rodrigo B. Abreu**[1], **Greg A. Kirchenbaum**[1,2¤], **Giuseppe A. Sautto**[1], **Emily F. Clutter**[1], **Ted M. Ross**[1,2]*

**1** Center for Vaccines and Immunology, University of Georgia, Athens, Georgia, United States of America,
**2** Department of Infectious Diseases, University of Georgia, Athens, Georgia, United States of America

¤ Current address: Cellular Technology Limited (CTL), Shaker Heights, OH, United States of America
* tedross@uga.edu

**Data Availability Statement:** All relevant data are within the manuscript and its Supporting Information files.

## Abstract

Influenza is a highly contagious viral respiratory disease that affects million of people worldwide each year. Annual vaccination is recommended by the World Health Organization with the goal of reducing influenza severity and limiting transmission through elicitation of antibodies targeting the hemagglutinin (HA) glycoprotein. The antibody response elicited by current seasonal influenza virus vaccines is predominantly strain-specific, but pre-existing influenza virus immunity can greatly impact the serological antibody response to vaccination. However, it remains unclear how B cell memory is shaped by recurrent annual vaccination over the course of multiple seasons, especially in high-risk elderly populations. Here, we systematically profiled the B cell response in young adult (18–34 year old) and elderly (65+ year old) vaccine recipients that received annual split inactivated influenza virus vaccination for 3 consecutive seasons. Specifically, the antibody serological and memory B-cell compartments were profiled for reactivity against current and historical influenza A virus strains. Moreover, multiparametric analysis and antibody landscape profiling revealed a transient increase in strain-specific antibodies in the elderly, but with an impaired recall response of pre-existing memory B-cells, plasmablast (PB) differentiation and long-lasting serological changes. This study thoroughly profiles and compares the immune response to recurrent influenza virus vaccination in young and elderly participants unveiling the pitfalls of current influenza virus vaccines in high-risk populations.

## Introduction

Seasonal influenza virus infection remains a major public health concern with significant social and economic impact. During the 2018–2019 northern hemisphere influenza season, more than 30 million people were sick with influenza with >50% seeking healthcare services. Influenza is classified as of moderate severity disease by U.S. Centers for Disease Control and Prevention (CDC), with influenza viruses causing ~500,000 hospitalizations and ~30,000

**Funding:** This work was funded, in part, by the University of Georgia (UGA) (UGA-001) and the Emory-UGA Center of Excellence of Influenza research and Surveillance (Emory-UGA CEIRS) contract grant (HHSN272201400004C). The content is solely the responsibility of the authors and does not necessarily represent the official views of the NIH. In addition, TMR is supported by the Georgia Research Alliance as an Eminent Scholar. GAK's contributions to this manuscript preceded his current position at Cellular Technology Limited (CTL), and CTL was not involved in any part of the study. The funding organizations did not play a role in the study design, data collection and analysis, decision to publish, or preparation of the manuscript.

**Competing interests:** GAK is currently affiliated with Cellular Technology Limited (CTL) but his contributions to this manuscript preceded his current position at CTL, and CTL had no involvement in this work. This does not alter our adherence to PLOS ONE policies on sharing data and materials.

deaths annually [1]. The World Health Organization (WHO) recommends annual vaccination to prevent seasonal influenza virus infection and transmission. Nonetheless, vaccination effectiveness is low (generally below 50%) and highly variable between influenza virus subtypes. In the U.S. each year, ≅50% of the population is vaccinated each season, which is far from the Healthy People 2020 goal of 70% coverage [2]. This results in a large proportion of the population at risk of influenza virus infection each year [3]. Furthermore, the young and the old are disproportionately impacted by influenza virus induced disease [4], with vaccinations having a lower effectiveness in these high-risk populations [5].

Influenza viruses undergo change (drift) from season to season forcing continued updating of the vaccine to include novel seasonal antigenic variants [6]. The current quadrivalent, inactivated influenza virus vaccines (QIV) mainly induce humoral immune responses, eliciting strain-specific receptor-blocking antibodies with a narrow breadth of neutralizing activity [7]. Recently, it was reported that there is antigenic competition between the four vaccine strains included in QIV, leading to a subdominant H3N2 immune response during the 2016–2017 influenza season [8]. Previous studies have hinted at the possibility of skewed immune responses to influenza virus vaccination as a consequence of past influenza virus exposures. Early-life exposure is generally described as original antigenic sin or imprinting [9–11]. This is particularly evident when comparing the responses to H1N1 and H3N2 influenza A viruses (IAV) in participants born when only one of these subtypes circulated in the human population [8, 12–14]. However, the impact of recurrent vaccination on the immune response to QIV remains controversial [15–17]. In elderly populations, recurrent vaccination was first reported to enhance neutralizing antibodies to influenza B viruses [18]. In contrast, recurrent influenza virus vaccination might hinder neutralizing antibody responses and decrease vaccine effectiveness, particularly against H3N2 IAV strains [16].

Inactivated influenza virus vaccines are poorly immunogenic and mainly rely on pre-existing immune memory. Recent developments in single cell sequencing technologies have begun to unravel the complex process of memory B-cell (Bmem) recall, clonal expansion, affinity maturation and plasmablast (PB) expansion that follows influenza virus vaccination [19–22]. Still, it remains unclear how recurrent influenza virus vaccination shapes the memory B-cell compartment and the influenza-reactive serological antibody profile. In this report, the composition of serum and B cell memory polyclonal antibodies in young adult and elderly participants was tracked following recurrent vaccination for three consecutive influenza seasons. Through systematic analysis of serological antibody binding and hemagglutination-inhibition (HAI) activity against current and past influenza strains, the polyclonal antibody reactivity in young and elderly participants was profiled. Furthermore, unlike young adults, elderly participants have transient rises in antibody with HAI activity to the current influenza strains, but with minimal long-term changes in the influenza-reactive antibody profile. Mechanistically, these seems to be associated with inefficient differentiation of pre-existing vaccine-reactive Bmems into antibody-secreting PB following influenza virus vaccination.

## Results

### Recurrent vaccination redirects serological repertoire to receptor-blocking antibodies

Immunological changes following influenza virus vaccination are generally assessed through serological hemagglutination inhibition (HAI) activity as a surrogate of receptor-blocking antibodies [23–25]. Influenza virus vaccines strongly induce receptor-blocking antibodies in healthy young adults, but less efficiently in the elderly [24, 26, 27]. In contrast, the long-term impact of vaccination, particularly in the context of recurrent influenza virus vaccination is

controversial and less-well understood [16, 18, 28]. To measure changes in the serological antibody response to yearly recurrent influenza virus vaccination in the young and elderly, 50 participants (16 young adults and 34 elderly) were vaccinated in 3 consecutive influenza seasons (2014–2015 through 2016–2017) with the split inactivated influenza virus vaccine (Fluzone®, Sanofi Pasteur, Swiftwater, PA, USA). Sera was collected pre- and post-vaccination and anti-HA specific antibodies were measured against each of the HA vaccine components. In addition, the HAI titers against both H1N1 and H3N2 vaccine components were assessed (Fig 1A). With a biparametric analysis of the anti-HA elicited antibodies, participants were categorized as high-HAI (Q1), high-non-HAI (Q2), strong-HAI (Q3) serological profiles against both IAV vaccine components. Influenza virus vaccination efficiently elicited HAI activity ~28 days post-vaccination in young and elderly participants (S1 Fig). In young adult participants, recurrent vaccination with the exact same vaccine strain (*i.e.* H1N1) induced long-term persistent changes in the serological profile towards receptor-binding epitopes ($\chi^2$ 2016D0_2014D0 *p = 0.007*, Fig 1B). In contrast, elderly participants had a transient increase in HAI activity, but insignificant long-term changes in the overall antibody profile ($\chi^2$ 2016D0_2014D0 *p = 0.19*, Fig 1B). In parallel, recurrent vaccination with antigenically distinct strains resulted in ≅45% of young participants acquiring serological cross-reactive HAI activity to the new H3N2 vaccine strain in 2015 prior to vaccination and this number increased to greater than 65% in 2016 (Fig 1C). Again, elderly participants can transiently adapt their serological antibody repertoires to the new H3N2 antigenically drifted HA proteins, but only 45% of elderly participants had cross-reactive HAI activity to the new H3N2 vaccine strain in 2016 (Fig 1C).

## Subdominant response to H3N2 vaccine component

Previously, we reported a significant subdominant immune response to the H3N2 vaccine component in 2016 [8]. To understand the impact of recurrent influenza vaccination on the immunogenicity of individual H3N2 vaccine strains, the percentage of anti-H3 binding antibodies was compared to the total response against all the IAV vaccine components over the three consecutive seasons (Fig 2A–2D). Young participants had an overall balanced antibody response to both H1N1 and H3N2 vaccine components in 2014 and 2015 (Fig 2A and 2B). In contrast, elderly participants had a significantly subdominant antibody response to the H3N2 vaccine component (% H3 binding $\neq$ 50% *p<0.0001*) (Fig 2A and 2B). As previously reported [8, 17], pre-immunity against the H3N2 vaccine component in 2016 was significantly subdominant compared to the humoral response against the H1N1 vaccine component in both young and elderly participants and did not change following influenza virus vaccination (Fig 2C and 2D). Interestingly, there were overall balanced HAI activity against H3N2 and H1N1 vaccine components in both young and elderly participants to both IAV vaccine components.

## Vaccination elicits a vaccine-specific plasmablast response

Vaccine-induced changes in the serological antibody repertoire derive from PB expansion and differentiation following vaccination [29]. To understand why the elderly had such transient changes in theirs serological repertoires compared to young participants, we selected 12 participants (6 young and 6 elderly) and quantified the frequency of PBs (CD27+/CD38++/CD20-) in peripheral blood 7 days after influenza virus vaccination (Fig 3A). Young adult participants had a prominent increase in B-cell PBs every year following vaccination (Figs 3B and S2). In contrast, elderly participants exhibited a minimal increase in the frequency of circulating PBs 7 days post-vaccination. Interestingly, 3 of the 6 elderly participants analyzed possessed elevated frequencies of circulating PBs across the multiple time-points, regardless of vaccination (D#1089, D#1132 and D#1132) (Fig 3C).

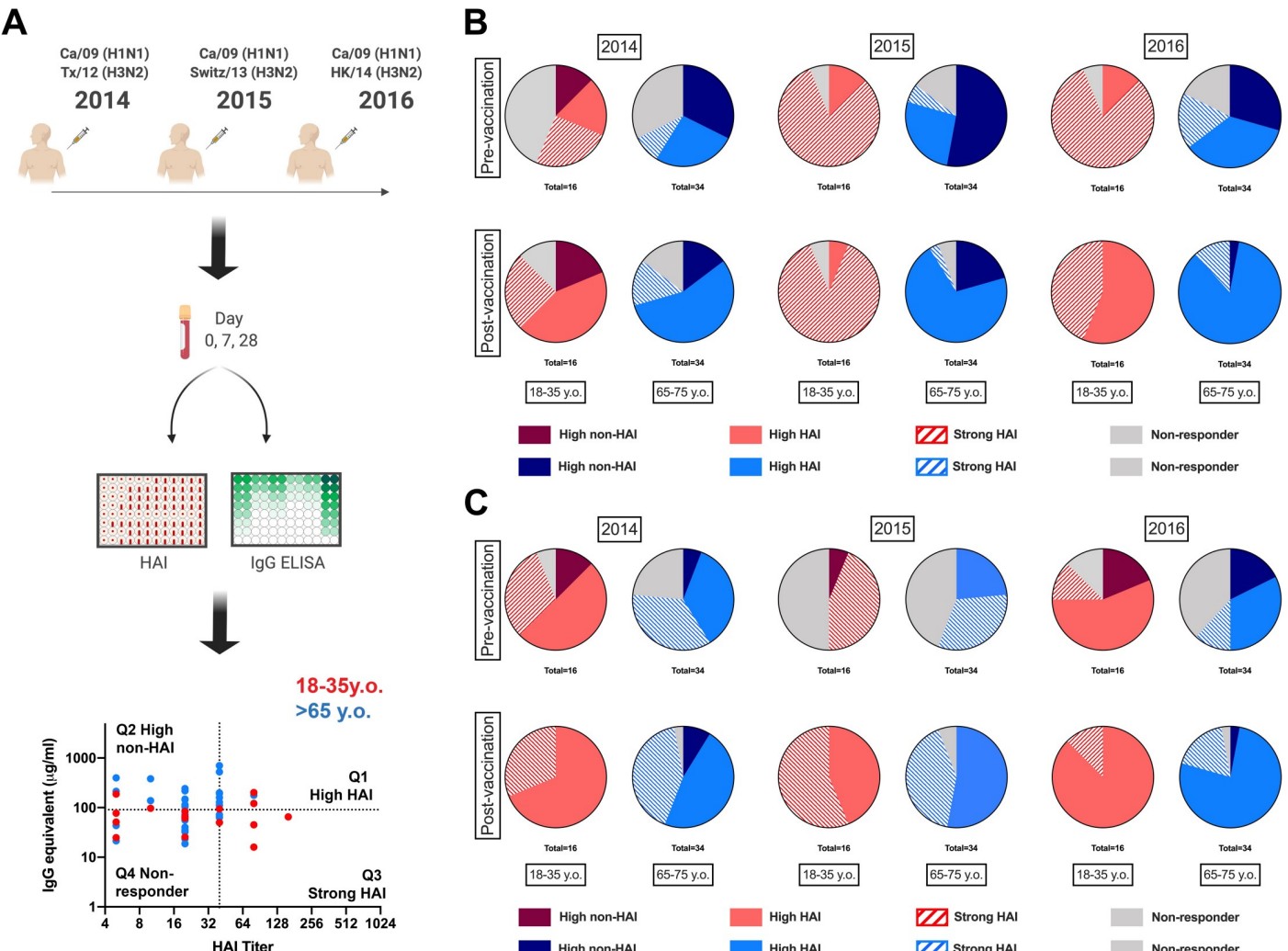

**Fig 1. Changes in serological antibody profile following recurrent influenza vaccination.** A) General experimental approach for serological profiling. 50 participants (16 young-adult and 34 elderly) were vaccinated for three consecutive years with standard of care inactivated influenza vaccine and serum samples collected prior to and 21–28 days post-vaccination. Serum samples were tested for hemagglutination inhibition (HAI) activity against the H1N1 and H3N2 vaccine virus strains as described in M&M section. In parallel total rHA-reactive IgG-antibodies were quantified by ELISA as described in the M&M analysis. Biparametric quadrant analysis of each subject's HAI titer and rHA-specific IgG (μg/ml) identified participants with High-HAI antibodies in Q1, high non-HAI in Q2, strong-HAI in Q3 and non-responders in Q4. B) Changes in H1N1-reative serological antibodies in young-adult (red) and elderly (blue) participants vaccinated for three consecutive years, measured as in A. C) Changes in H3N12-reative serological antibodies in young-adult (red) and elderly (blue) participants vaccinated for three consecutive years, measured as in A. Changes in the proportion of participants in each quadrant over time were assess by Chi-square test ($\chi^2$).

To assess the frequency of antigen-specific PBs in circulation 7 days after vaccination, PBMCs were stained with fluorochrome-conjugated rHA probes, as previously described [8, 30]. Furthermore, to distinguish vaccine-specific from past cross-reactive PB responses, PBMCs were stained with both the current vaccine and a pool of historical influenza strains rHA probes (Fig 3A). Both H1N1 and H3N2 PB responses are highly vaccine-specific (S3 Fig) in young and elderly participants, while the frequency of broadly-reactive PBs is generally much lower (except D#1089 and D#1108). As extensively reported previously, influenza virus vaccination fails to recall historical-specific (and non-cross-reactive) PB responses. The serological responses to H3N2 influenza vaccine component are highly subdominant compared to H1N1 vaccine strain (Fig 2). When the frequencies of vaccine-specific PBs against H1N1 and

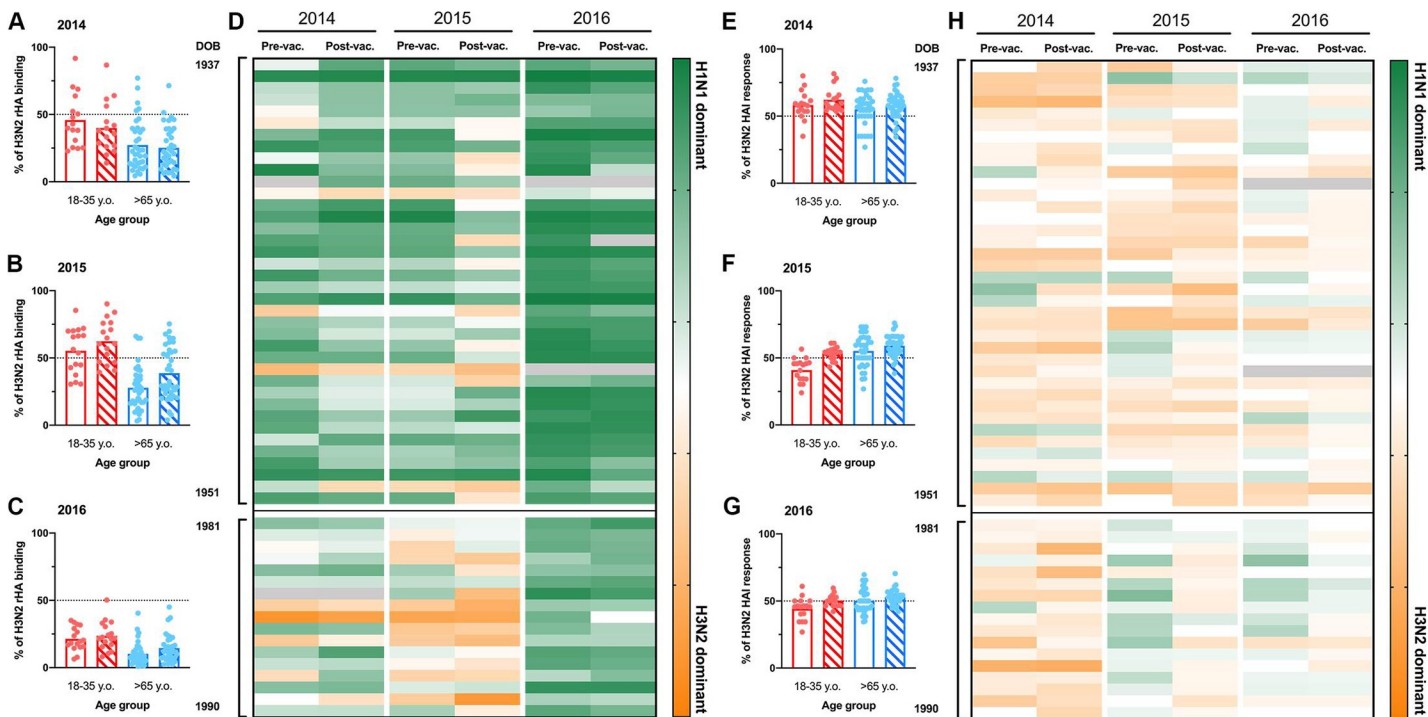

**Fig 2. H3N2 vaccine immunogenicity during recurrent vaccination.** A-C) Percentage of H3N2 rHA-binding relative to the total serum IgG antibodies against IAV vaccine strains of 50 participants (16 young adult and 36 elderly) in 2014 (A), 2015 (B) and 2016 (C). The dashed-line represents the hypothetical balanced response to both IAV (H1N1 and H3N2) vaccine components. D) IAV subtype immunodominance, based in total rHA-reactive antibodies, in 50 participants organized by age (oldest to youngest from top to bottom) vaccinated for three consecutive years. Dark green represents 100% rHA binding to H1N1 vaccine component. Bright orange represents 100% rHA binding to H3N2 vaccine component. E-G) Percentage of serum HAI activity against the H3N2 vaccine strain relative to total serological HAI activity against IAV vaccine components of 50 participants (16 young adult and 36 elderly) in 2014 (E), 2015 (F) and 2016 (G). The dashed-line represents the hypothetical balanced response to both IAV (H1N1 and H3N2) vaccine components. D) IAV subtype immunodominance, based in serological HAI activity, in 50 participants organized by age (oldest to youngest from top to bottom) vaccinated for three consecutive years. Dark green represents 100% HAI titer response to H1N1 vaccine component. Bright orange represents 100% HAI titer responses to H3N2 vaccine component. Gray boxes represent missing values.

H3N2 vaccine components were assessed, again a subdominant PB response was observed against the H3N2 vaccine component in young and elderly participants (Figs 3D, 3E and S3).

### Young adult serological profile in response to recurrent influenza vaccination

The impact of past influenza virus exposures on the immune response to influenza virus vaccination has been previously reported [8, 12–14]. To profile the serological antibody repertoire, we measured the levels of antibodies reactive against a panel of historical rHAs in 6 young adult participants (Figs 4 and S4). In parallel, to profile the specificity of HAI activity, serological HAI activity was screened against an extensive panel of current and historical IAV. While some participants (D#1008 and D#1032) show signs of imprinting against both H1N1 and H3N2 strains (Figs 4A–4D and S4) from mid to late 1990's, other participants (D#1011 ad D#1137) had signs of imprinting to just one of the IAV subtypes (Figs 4E, 4F and S4). Finally, one last young subject had signs of imprinting to an H3N2 strain from the late 1990's (S4 Fig).

Overall, vaccination does not drastically change the serological antibody repertoire (Fig 4A–4F). To understand which antibodies are recalled and adapted towards the vaccine strain, the rise in antibody levels was calculated (ΔD21-D0) against the vaccine strains and historical influenza strains 21–28 days after vaccination, over three consecutive years (Fig 4G–4L).

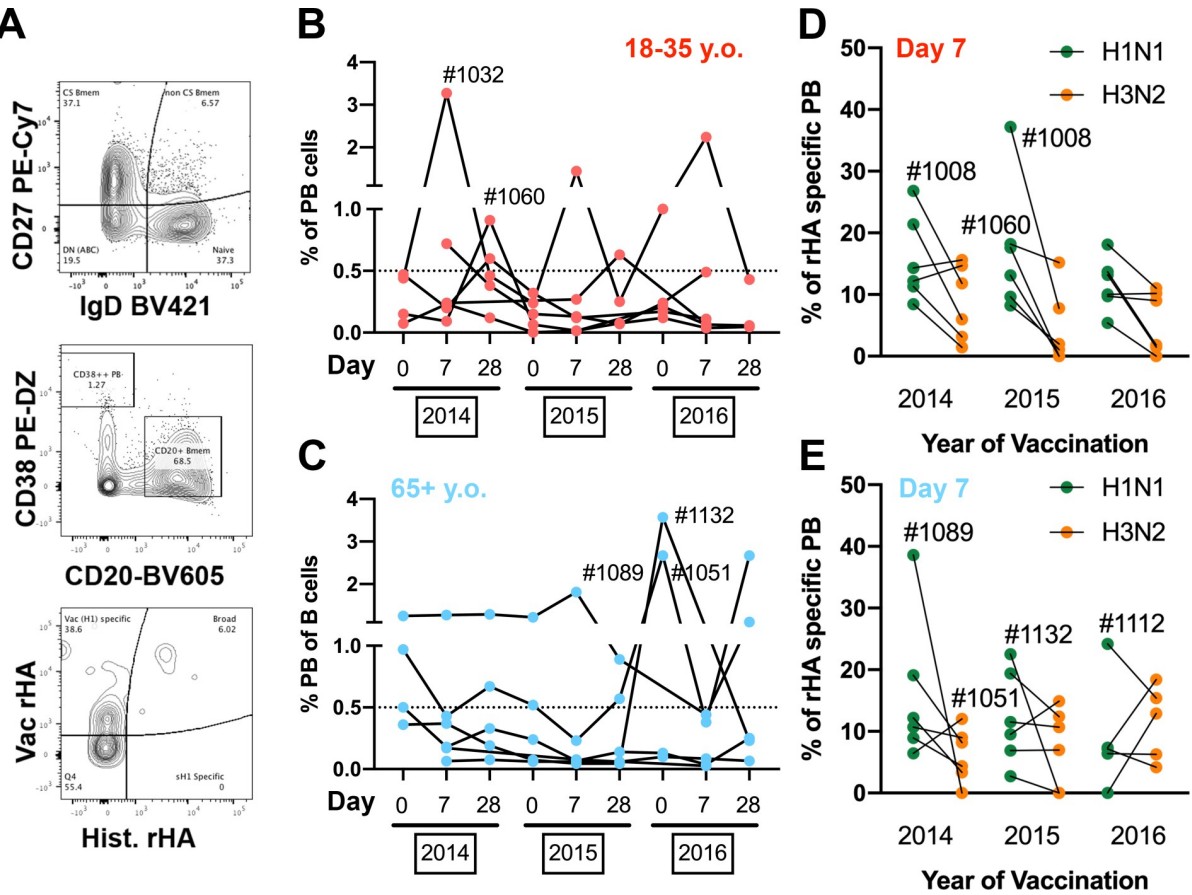

**Fig 3. Plasmablast response in young and elderly participants vaccinated for three consecutive years.** A) Representative gating strategy to quantify rHA-specific plasmablast (CD27⁺/CD38⁺⁺/CD20⁻) in the peripheral blood of vaccinated participants. B-C) Changes in frequency of plasmablast B-cells in the peripheral blood of young-adult (B) and elderly (C) participants vaccinated over three consecutive years. D-E) Frequency of vaccine-specific (H1N1 in green and H3N2 in orange) plasmablast 7 days post-vaccination in young adult (D) and elderly (E) participants vaccinated for three consecutive years. Participants of interest are identified with the corresponding ID numbers. H1N1 Vac rHA is CA/09 for 2014–16, and H1N1 Hist. rHA are NC/99 and Sing/86 (pooled at half concentration); H3N2 Vac rHA is TX/12 for 2014, Switz/ 13 for 2015 and HK/14 for 2016; H3N2 Hist rHA are Pan/99 and Wisc/05 for 2014–16.

Again, despite such close age-range, we observe tremendously different responses to influenza virus vaccination (Fig 4G–4J). Participants imprinted with both H1N1 and H3N2 influenza strains (D#1008 and 1032) had a moderate increase in cross-reactive antibodies against both current and historical IAV strains following vaccination in 2014 and 2015. Strikingly, in 2016, these two participants had contrasting responses to influenza virus vaccination. While D#1008 mainly recalled cross-reactive antibodies against the current and past H3N2 influenza strains, D#1032 demonstrated a pronounced decrease in their overall H3N2-reactive serological antibody repertoire (Fig 4G–4J). In parallel, a participant originally imprinted with an H3N2 IAV (D#1011) had a significant increase in their cross-reactive H1N1 antibody repertoire following the first vaccination in 2014 ($p = 0.032$), with continuous adaptation of their antibody repertoire to the H1N1 vaccine strain in subsequent seasons (Fig 4K). The anti-H3N2 immune response was also marked by a moderate increase in serological antibodies against both current and historical strains, without pronounced changes in the overall repertoire (Fig 4L). Finally, the impact of influenza virus vaccination on antibodies with HAI activity is diverse amongst young participants (S5 Fig). While some participants had a marked increase in

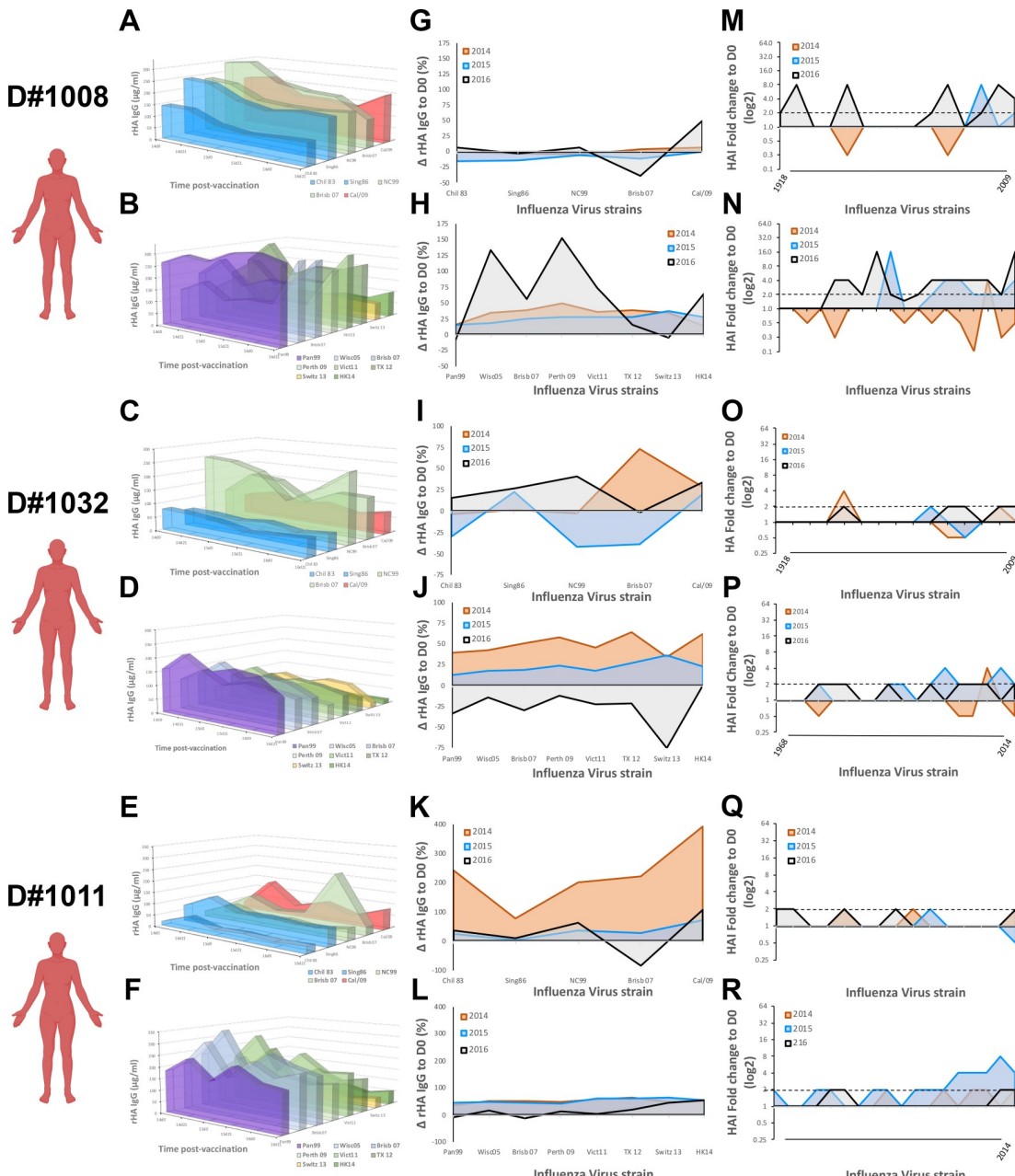

**Fig 4. Serological antibody landscape in young participants vaccinated for three consecutive years.** A,C,E) Serological IgG antibodies against rHA from current H1N1 vaccine strain and 4 historical seasonal H1N1 virus strains (1983–2007) in three young participants vaccinated for three consecutive years. Colors represent antigenically similarity between H1 rHA. B,D,F) Serological IgG antibody levels against rHA the current H3N2 vaccine strains and 5 historical seasonal H3N2 virus strains (1999–2011) in three young participants vaccinated for three consecutive years. Colors represent antigenically similarity between H3 rHA. G,I,K) Changes in serological antibody levels against rHA from different H1N1 virus strains, measured as in A, 21–28 days after vaccination in young participants vaccinated for three consecutive years. H,J,L) Changes in serological antibody levels against rHA from different H3N2 virus strains, measured as in B, 21–28 days after vaccination in young participants vaccinated for three consecutive years. M,O,Q) Changes in serological HAI activity titer against different H1N1 virus strains (1918–2009) 21–28 days after vaccination in young participants vaccinated for three consecutive years. N,P,R) Changes in serological HAI activity titer against different H3N2 virus strains (1968–2016) 21–28 days after vaccination in young participants vaccinated for three consecutive years.

broadly-reactive antibodies with HAI activity (Fig 4M and 4N), others showed minimal changes in their HAI activity, even against the vaccine strain (Fig 4O and 4R).

## Elderly participants show transient rises in HA-specific antibody titer

Elderly participants had impaired PB responses and long-term adaption of their serological antibody repertoire to the "new" drifted vaccine strains (Figs 1–3). To profile the serological antibody repertoire in 6 elderly participants, antibody titers were measured for reactivity against a panel of historical (1980–2016) rHAs (Figs 5 and S6). Elderly participants have similar levels of rHA-specific antibodies against current and past IAV strains; however, extremely polarized serological signatures were also observed, generally directed against H1N1 (Figs 5A–5F and S6). To assess imprinting in elderly participants, serological HAI activity was screened against an extensive panel of current and historical IAV strains (1918–2016) (S7 Fig). The oldest participants tested were born in 1934 (D#1089) and 1937 (D#1132). Each of these participants had increased HAI titers against H1N1 strains that were circulating in the 1940s (S7 Fig). Interestingly, the remaining elderly participants that were born in 1939 and 1940, had high HAI activity against H1N1 strains from 1940's, but also higher HAI titers against H1N1 and H3N2 IAV strains from 1970–1980's (S7 Fig).

To understand if pre-existing antibody immunity is recalled and adapted towards the vaccine strain, the rise in antibody titers (ΔD21-D0) was calculated against the vaccine strains and historical influenza strains 21 days after vaccination, over three consecutive seasons (Fig 5G–5L). Despite a prominent increase in antibody levels with broad cross-reactivity following vaccination (Fig 5G–5L), these waned significantly by the subsequent year (S6 Fig). In the elderly, this transient increase in HA-specific antibodies often translated into increased serological HAI activity against current and historical IAV strains (Figs 5M–5R and S7), particularly after three consecutive vaccinations.

## Impaired memory recall responses in the elderly

Inactivated influenza virus vaccines heavily rely on memory B-cell (Bmem) responses for protective immunity [31]. To measure the impact of recurrent influenza virus vaccination on the memory B-cell compartment in young and elderly participants, the frequency of HA-specific amongst class-switched memory B-cells (CS-Bmems) was tracked by flow cytometry (Fig 6A). Overall, the frequency of historical-reactive memory B-cells was higher than vaccine-specific (Fig 6B). However, when tracking the dynamics of HA-specific CS-Bmems over time, there was an increase in vaccine-specific CS-Bmems relative to cells reactive to historical strains 7–9 days after vaccination (Fig 6C and 6D). In contrast, recurrent influenza virus vaccination had minimal impact on broadly-reactive CS-Bmems (Fig 6E).

Influenza virus vaccination elicits subdominant PB and serological H3N2 responses compared to immune responses against the H1N1 vaccine component (Fig 2). Similarly, the frequency of H3N2-reactive CS-Bmems was lower than the corresponding H1N1-reactive compartment (Fig 6F and 6H). Despite their low frequency, broadly-reactive CS-Bmems were well-balanced between H1N1 and H3N2 IAV strains.

Overall, there was no difference in the frequency of vaccine-specific CS-Bmems in young and elderly participants. To ascertain the potential and binding profile of Bmem-derived antibodies, unfractionated PBMC were subjected to *in vitro* differentiation and conditioned supernatant samples were screened for reactivity against a panel of rHA proteins representing current and past IAV strains (Figs 6I, S8 and S9). Each season, young participants had increased HA-specific Bmem-derived antibodies 21–28 days post-vaccination. Interestingly, most young participants had a Bmem-derived antibody repertoire highly skewed towards the

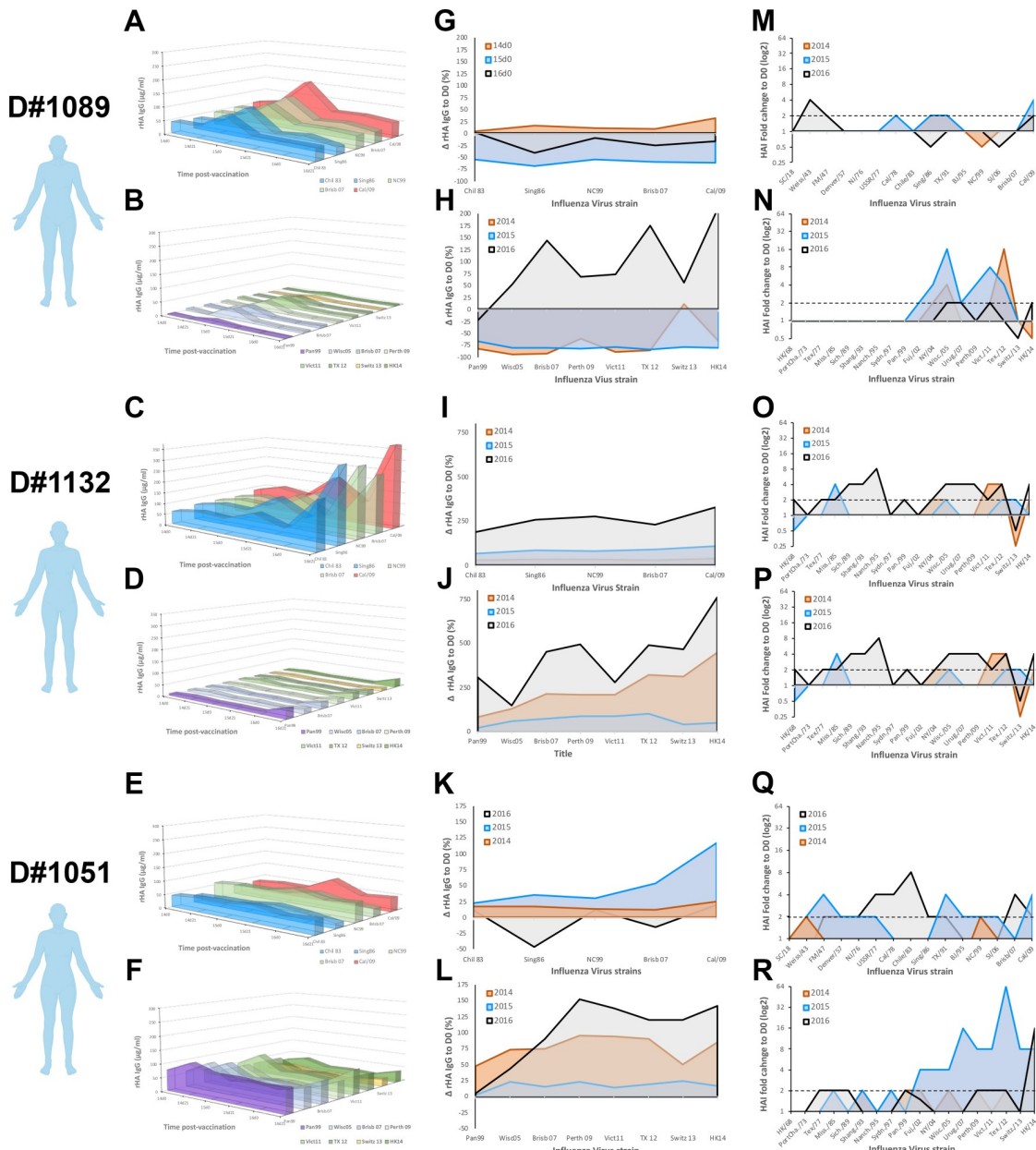

**Fig 5. Serological antibody landscape in elderly participants vaccinated for three consecutive years.** A,C,E) Serological IgG antibodies against rHA from current H1N1 vaccine strain and 4 historical seasonal H1N1 virus strains (1983–2007) in three elderly participants vaccinated for three consecutive years. Colors represent antigenically similarity between H1 rHA. B,D,F) Serological IgG antibody levels against rHA the current H3N2 vaccine strains and 5 historical seasonal H3N2 virus strains (1999–2011) in three elderly participants vaccinated for three consecutive years. Colors represent antigenically similarity between H3 rHA. G,I,K) Changes in serological antibody levels against rHA from different H1N1 virus strains, measured as in A, 21–28 days after vaccination in elderly participants vaccinated for three consecutive years. H,J,L) Changes in serological antibody levels against rHA from different H3N2 virus strains, measured as in B, 21–28 days after vaccination in elderly participants vaccinated for three consecutive years. M,O,Q) Changes in serological HAI activity titer against different H1N1 virus strains (1918–2009) 21–28 days after vaccination in elderly participants vaccinated for three consecutive years. N,P,R) Changes in serological HAI activity titer against different H3N2 virus strains (1968–2016) 21–28 days after vaccination in elderly participants vaccinated for three consecutive years.

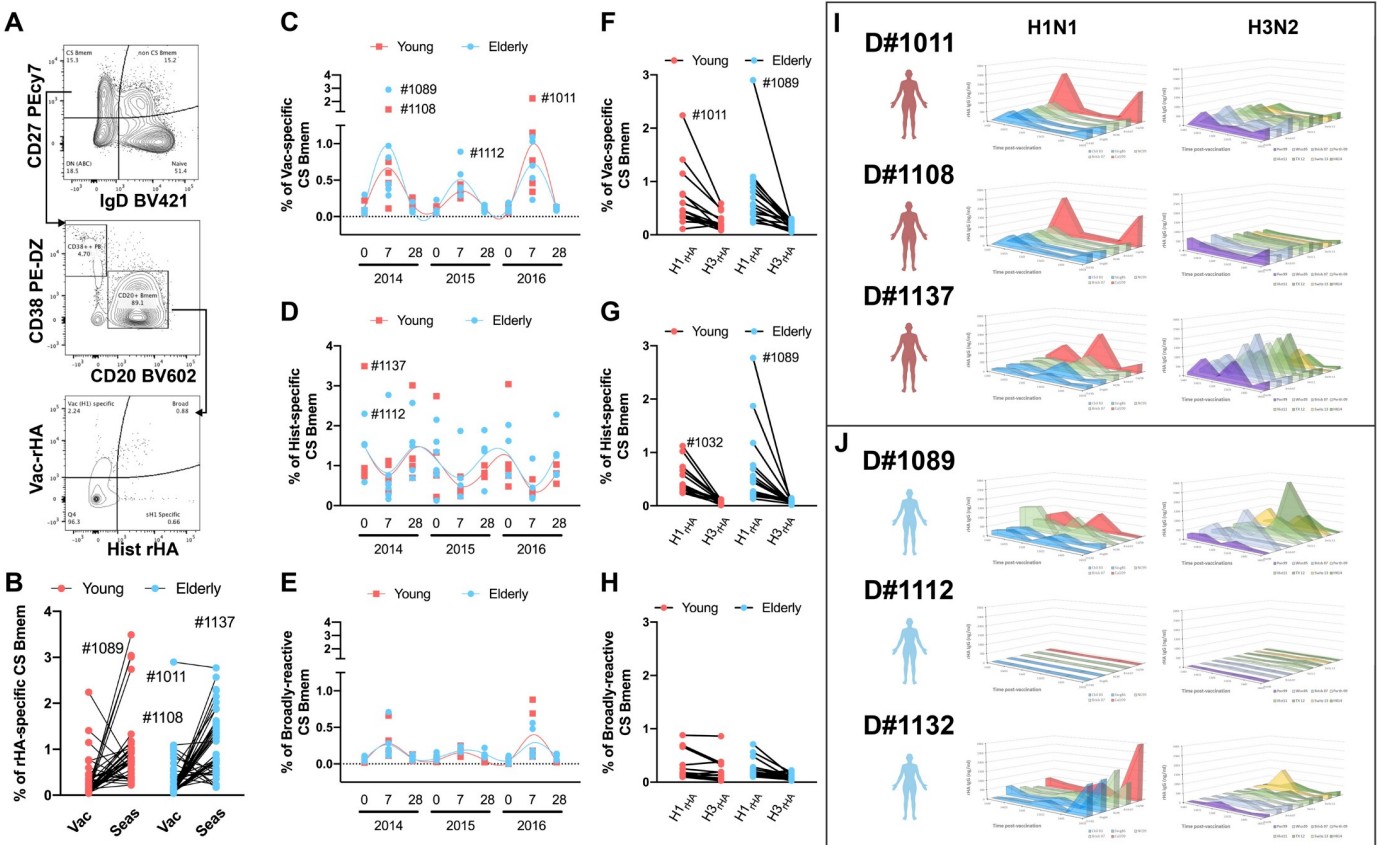

**Fig 6. Class-switched memory B-cell (CS-Bmem) responses in young and elderly participants vaccinated for three consecutive years.** A) Representative gating strategy to quantify rHA-specific CS-Bmem in the peripheral blood of vaccinated participants. B) Percentage of vaccine-specific and historical-specific CS-Bmem in the peripheral blood of young (red) and elderly (blue) vaccinated participants. C-E) Percentage of vaccine-specific (C), historical-specific (D) and broadly-reactive (E) CS-Bmem in young (red) and elderly (blue) participants vaccinated for three consecutive years. Lines show cubic spline interpolation model for young adult (red) and elderly (blue) participants. F-H) Percentage of H1N1 and H3N2 vaccine-specific (F), historical-specific (G), and broadly reactive (H) CS-Bmem in young adult(red) and elderly participants vaccinated for three consecutive years I) Bmem-derived IgG antibodies against rHA from current vaccine strain and historical seasonal virus strains, as in Figs 4 and 5, in three young adult (red) and three elderly (blue) participants vaccinated for three consecutive years.

recent H1N1 vaccine strains with decreased reactivity against historical strains (Figs 6I and S8). In contrast, H3N2 Bmem-derived antibodies were generally lower and exhibited a broader binding profile (Figs 6I and S8).

Half of the tested elderly participants had no increase in Bmem-derived antibodies after vaccination (Figs 6J and S9). The oldest participants tested (D#1089) showed a significant increase in Bmem-derived antibodies 21–28 days after vaccination in 2014 and 2015. However, in this subject, Bmem-derived antibodies had higher reactivity against historical IAV than the current vaccine strain (Fig 6J). Furthermore, the prominent increase in Bmem-derived antibodies was followed by an almost complete depletion of the rHA-reactive Bmem compartment in 2016. In contrast, two other elderly participants had a significant rise in Bmem-derived antibodies in 2016, occurring after their third consecutive vaccination (Figs 6H and S9). Noticeably, both participants had an increase in Bmem-derived antibodies with broad reactivity against the H1N1 historical viruses, but with higher specificity towards the vaccine strain. The opposite was observed in regards to the immune responses against the H3N2 vaccine component, with an increase in their reactivity towards historical H3N2 IAV strains (Figs 6H and S9).

## Elderly participants have increased frequencies of vaccine-specific atypical B-cells

Recent reports exposed the biological relevance of double negative (CD27-/IgD-) B-cells (DN-C) [32–35]. To measure the impact of influenza virus vaccination on the DN-C compartment in young and elderly participants, changes in the frequency of HA-specific DN-C were tracked by flow cytometry after staining with rHA-probes from past and current IAV strains. Overall, the dynamics of vaccine-specific DN-C is similar to that of CS-Bmems with an increase 7 days post-vaccination and concurrent with a decrease in historical IAV HA-specific DN-C (Fig 7A and 7B). However, particularly during the 2014 and 2015 influenza seasons, elderly participants had higher frequencies of vaccine-specific DN-C than young adult participants (Fig 7A). Also, in this case, the frequency of broadly-reactive DN-C is lower (Fig 7C). Moreover, the frequency of H1N1 vaccine and historical HA-specific DN-C is generally higher than the frequency of cognate CS-Bmems, especially in the elderly (Fig 7E and 7F). Interestingly, despite the lower frequencies of H3N2 rHA-B-cells, these seem to be equally distributed between the CS-Bmem and DN-C compartments (Fig 7E and 7F). Finally, the frequency of H1N1 and H3N2 broadly-reactive B-cells was similarly low in both CS-Bmem and DN-C compartments (Fig 7G).

## Discussion

Shortly after Wilson Smith and colleagues first identified the etiological agent of influenza disease in 1933 and proved the induction of strong neutralizing humoral responses following influenza A virus (IAV) infection [36], they quickly recognized that antigenically drifted strains could evade the host pre-immunity and cause subsequent infections [37]. Since then, over 50 different IAV strains have been used in vaccine formulations to control seasonal influenza virus epidemics [38]. Over the past ten years the H3N2 vaccine strain included in the northern hemisphere has been were updated 7 times, while during this same interval the H1N1 vaccine strain component, representing the swine-origin influenza virus (SOIV) causing the 2009 pandemic, was not updated until the 2016–2017 season. In parallel, the past decade was marked by extensive efforts to increase vaccine coverage in high-risk populations, especially in infants and elderly subjects. Nonetheless, there is a need for comprehensive longitudinal studies assessing the impact of recurrent influenza virus vaccination in the elderly. While previous studies often lack in-depth immunological analyses, in this study the serological and memory B-cell responses were systemically characterized following recurrent influenza virus vaccination over three consecutive years, 2014–2015 to 2016–2017.

From October 2014 to March 2017, the U.S. experienced three influenza seasons of low to mild influenza virus activity. The 2014–2015 and 2016–2017 seasons were dominated by H3N2 influenza viruses with higher infection and hospitalization rates than the 2015–2016 season, which was dominated by H1N1 influenza viruses [39–41]. During these three seasons, the H3N2 vaccine strain was updated each year [42]. Vaccine effectiveness across all ages and against all vaccine strains ranged from 20 to 50%, with the lowest in 2014 against H3N2 influenza viruses (5%) and the highest in 2015 against H1N1 influenza viruses (45%) [39, 43–45]. In high-risk populations, such as the elderly, vaccine effectiveness was higher than the overall average against H1N1 IAV in 2015, and consistently lower against H3N2 IAV in 2014 and 2016. Interestingly, the immediate subsequent influenza season (2017–2018) was marked by an extremely severe disease outcome [45], even in the absence of significant antigenic drift between concurrent vaccines strains [42].

Our group described the impact of influenza virus vaccination between 2013 to 2017 on IAV vaccine-specific serological antibodies across different ages groups [24]. During this

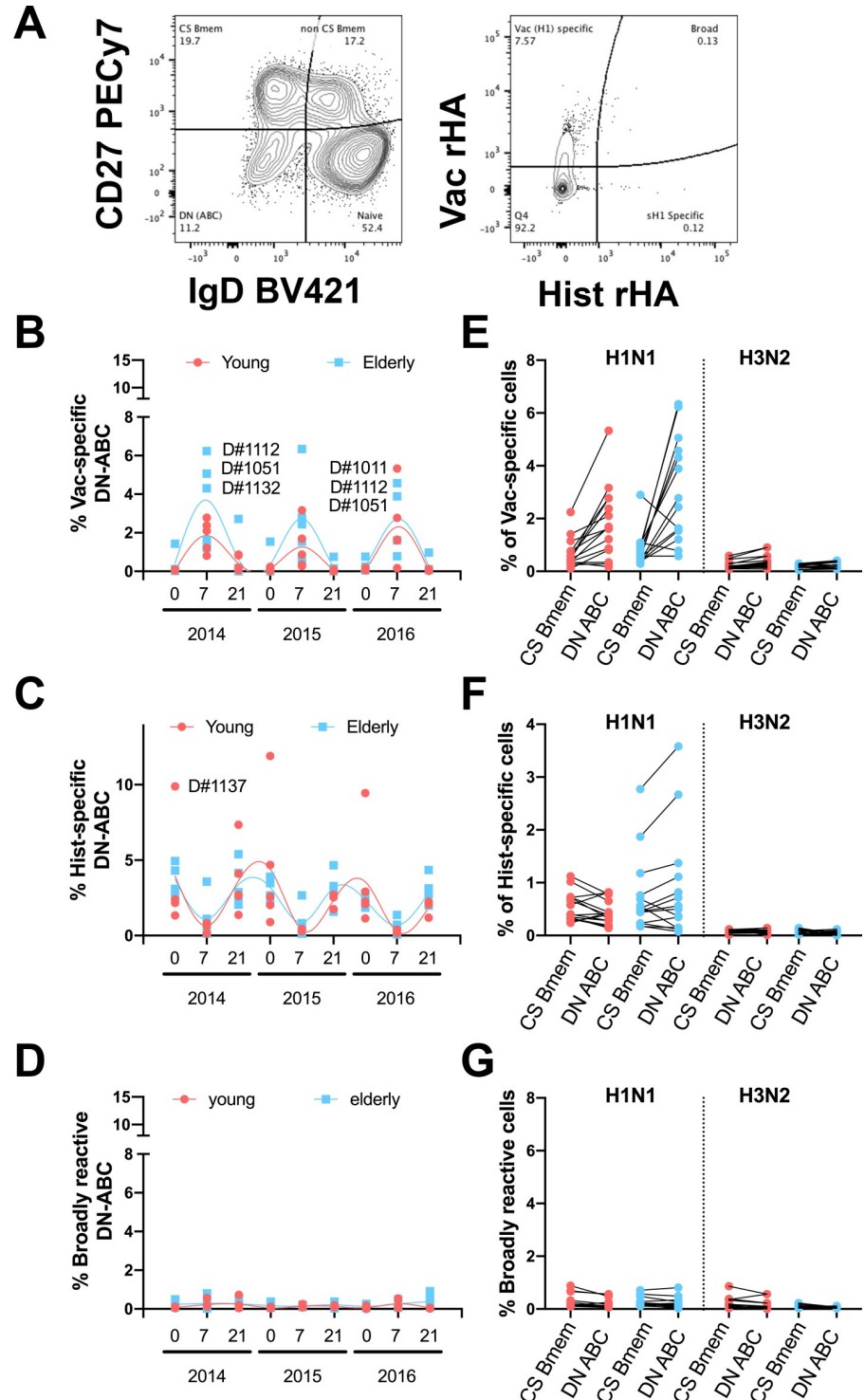

**Fig 7. Frequency of rHA-reactive DN-C in young and elderly participants.** A) Representative gating strategy to quantify rHA-specific DN-C in the peripheral blood of vaccinated participants. B-D) Percentage of vaccine-specific (B), historical-specific (C) and broadly reactive (D) DN-C in young adult (red) and elderly (blue) participants vaccinated for three consecutive years. E-G) Frequency of H1N1 and H3N2 vaccine-specific (E), historical-specific (F), and broadly reactive (G) B-cells in the CS-Bmem or DN-C compartment in young adult and elderly participants 7 days after vaccination.

period, influenza virus vaccination elicited vaccine-specific neutralizing antibodies in 18–85 year old participants and back-boosted cross-reactive neutralizing antibodies to historical IAV strains from the past 30 years [24]. This was particularly evident in participants born after 1975, when both H1N1 and H3N2 IAV strains were circulating in the human population. At the time, it was not clear if this reflected a change in the influenza-reactive antibody repertoire or the result of continuous increases in influenza-specific antibody titers. Comparing serological HAI activity with total HA-reactive IgG antibodies over three consecutive seasons showed that recurrent vaccination redirects the serological influenza-reactive antibody repertoires towards antigenic sites involved in receptor binding. These changes were retained up to three seasons in young adults, but not in the elderly participants (Fig 1). Yearly updates with antigenically distinct vaccine strains required adaptation of the antibody profile to drifted epitopes, but again these changes persisted for more than a year in young adults (Fig 1C).

Recent studies explored the impact of influenza virus vaccination on the PB and memory B-cell repertoires through single-cell next generation sequencing. Despite the small subset of donors, these previous reports showed highly oligoclonal responses that originated from expansion of pre-existing memory B-cells [21]. In parallel, vaccination with dramatically different influenza virus vaccine strains requires adaptation through somatic hypermutation and affinity maturation of pre-existing memory B-cells in young adults. Furthermore, this process seems to be impaired in the elderly, but the mechanism is still not well understood [46]. Here, elderly participants showed impaired PB expansion following influenza virus vaccination (Fig 3). In contrast, young adults had increased vaccine-specific PB responses. Interestingly, elderly participants with abnormally high PB frequencies prior to vaccination showed stronger vaccine-specific PB responses and stronger serological changes following influenza virus vaccination (Figs 3 and 5). Unfortunately, no further information regarding participant chronic or acute inflammatory status is available, but this observation is aligned with the recent theory of "inflammaging" and how it can impact the response to infectious agents in the elderly [47–49]. Many have speculated that immunosenescence and impaired innate immune responses are the main reasons behind decreased vaccine effectiveness in the elderly [50]. It is therefore possible that elderly participants with chronic or acute inflammatory diseases at the time of vaccination may have improved immune responses to influenza virus vaccination as observed in this study.

B-cell repertoire single-cell sequencing or serum Ig-seq elegantly depicts the molecular evolution of the immune response to influenza virus vaccination or infection [20, 22, 51–55], but fails to capture the complex synergistic and competitive interactions of a polyclonal antibody mixture. Vaccination does not drastically change the influenza virus-reactive antibody landscapes (Figs 4 and 5). Young adults in this study have significantly higher antibody reactivity against historical IAV strains in circulation during the first ten years of life (1990's), reflective of original antigenic sin. In contrast, elderly participants have similar antibody titers against H1N1 influenza virus strains isolated in 1990's to present (Fig 5). Obviously, these IAV strains are not representative of those in circulation during early-life of elderly participants, however the significant H3N2 subdominant antibody profile in the elderly is likely a result of early-life exposure to H1N1 IAV strains (Figs 2, 3 and 5). This phenomenon is further supported by increased serological activity against H1N1 IAV strains in circulation in the 1940's (S7 Fig). Moreover, recurrent influenza virus vaccination significantly back-boosted the serological response against historical vaccine strains, but these responses were tremendously variable even between individuals born in the same year and a with similar pre-immunity background (Fig 4). While speculative, this is likely the result of divergent Bmem-repertoire recall following vaccination (Figs 6 and 7).

Since the first identification of the two separate lymphocyte lineages (B and T-cells) in 1965 [56], many new discoveries have clarified aspects of B-cell development and different activation stages [57]. The mechanism of T-cell interactions, memory B-cell differentiation, GC-reactions, and affinity maturation are now well understood [58–61]. In contrast, peripheral Bmem-cell fate is less clear and the intrinsic signals determining Bmem cell proliferation, longevity and PB differentiation are still convoluted [62–64]. Moreover, recent studies in the elderly, shed light on the possibility that individuals with autoimmune or chronic infection/inflammation related disorders exhibited an exhausted B-cell compartment [34, 35, 65]. Half of the elderly participants tested in this study had negligible Bmem-derived antibody responses following influenza virus vaccination, despite similar frequencies of influenza virus-reactive memory B-cells in the periphery 7 and 21–28 days after vaccination. Vaccine-specific Bmem significantly increased 7–9 days after vaccination at the cost of recalling historical strain-reactive Bmem cells. This is most likely reflective of rapid recall and proliferation of circulating memory B-cells through direct cognate BCR-signaling and may be independent of GC reactions. Interestingly, aside from 7 days after vaccine administration, the frequency of historical-reactive Bmem cells were consistently higher than that of vaccine-reactive Bmem cells in both young and elderly participants (Fig 6). It is likely that the majority of vaccine-specific Bmem cells continue to differentiate into PBs throughout the course of the 7–10 days post-vaccination window. The fact that the PB compartment is highly enriched with vaccine-specific cells gives further supports to this hypothesis (Fig 3). Alternatively, these cells may migrate into the mucosa tissue and reside in situ awaiting cognate BCR stimulation upon subsequent infection. In this study, the distribution of IgA vs IgG Bmem cells was not assessed, but, in some participants, IgA$^+$ cells can represent up to 50% of the memory B-cell compartment (CD27$^+$/IgD$^-$) that are particularly prone to populate mucosal sites.

Class-switched memory B-cells have long been defined by the expression of CD27 and negative for the IgD surface markers [8, 66–68]. Recent reports highlighted the biological relevance of CD27$^-$/IgD$^-$ double negative B-cells cells (DN-C) due to their higher frequencies in elderly participants and patients affected by autoimmune disorders [33–35, 65, 69, 70]. First associated with an exhausted phenotype, this compartment is now characterized by tremendous heterogeneity [71]. Further resolution of this compartment by CD38, CD21, CD11c and T-bet transcription factor expression reveals three different memory B-cell fates: CD38$^+$/CD21$^+$ memory precursors; CD38-/CD21-/CD11$_c^+$/T-bet$^+$ extrafollicular precursors; and CD38$^-$/CD21$^-$/FcRL4/5$^+$ exhausted B-cells [32, 35, 70, 72–74]. Here, vaccine-specific DN-ABCs kinetics resembled those of CS-Bmem, with an increase 7–9 days post-vaccination followed by a decrease 28 days after vaccination. However, this increase is significantly higher in elderly participants vaccinated in 2014 and 2015 than young adults. This difference between elderly and young adult participants was no longer detectable in 2016 after three consecutive recurrent vaccinations (Fig 7).

Perhaps the main limitation of this study relies on its small sample size, preventing definitive conclusions based on robust statistical inference. Participant recruitment for multi-year-long longitudinal studies with sufficient sample for multiparametric immunological profiling is challenging, and with such small numbers, individual variation stands-out. Nonetheless, this and other reports seem to point towards a common trend; that despite all variation, young vaccinees seem to develop and adapt their repertoire to newly drifted strains, while elderly vaccinees, recall pre-existing non-neutralizing antibodies. This study deeply profiles the immune responses to influenza virus vaccination in young and elderly participants, but it has deeper immunological implications to other multivalent vaccines and vaccinations or infections in the context of pre-existing immunity. The here reported antigenic competition between influenza virus vaccine components may explain the incomplete effectiveness of other polyvalent

formulations, such as the pneumococcus vaccine. This study also highlights the relevant impact of pre-existing immunity in the response to subsequent influenza vaccinations. This is also likely to be pertinent for other infectious agents, such as the current SARS-CoV-2 pandemic which has motivated intense vaccine development efforts. In fact, it is important to recognize that any vaccine testing and assessments of efficacy performed using immunologically naïve models may not accurately reflect the magnitude and/or fine-specificity of the antibody responses elicited in humans endowed with pre-existing immunity against influenza or other coronaviruses.

## Materials and methods

### Study design

**Ethics statement and role of the funding source.**   The study procedures, informed consent, and data collection documents were reviewed and approved by the University of Georgia Institutional Review Board. Volunteers were recruited at medical facilities in two sites: Pittsburgh, Pennsylvania and Stuart, Florida. All were enrolled with written, informed consent.

The funding source had no role in sample collection, nor decision to submit the manuscript for publication.

**Participants and vaccine.**   Eligible volunteers between the ages of 18 to 35 and 65 to 85 years old (y.o.), who had not yet received the seasonal influenza vaccine, were enrolled beginning in September of each year, from 2014 to 2016. All vaccine formulations are based on World Health Organization recommendations for the Northern Hemisphere influenza seasons beginning in the Fall, and as such, all vaccinations and sample collections occurred each year between September 1$^{st}$ to December 15$^{th}$. Influenza virus did not circulate widely in the community during the time periods that the volunteers participated, and as such, participants were not monitored for influenza virus infection during that time-period; they were however asked during each visit if they had flu symptoms, and those who did were excluded from the study. Volunteers were recruited at medical facilities in two sites: Pittsburgh, Pennsylvania and Stuart, Florida. All were enrolled with written, informed consent. Exclusion criteria included documented contraindications to Guillain-Barré syndrome, dementia or Alzheimer disease, allergies to eggs or egg products, estimated life expectancy <2 years, medical treatment causing or diagnosis of an immunocompromising condition, or concurrent participation in another influenza vaccine research study. These two cohorts spanned for four years from 2013 to 2016 [24, 25]. However, for this study only the 50 (16 young and 34 elderly) repeatedly vaccinated participants from 2014 to 2016 were selected for serological antibody profiling. Serological hemagglutination inhibition (HAI) responses from recurrent vaccinated participants were similar to matching age groups of the original cohorts. Blood (70–90 mL) was collected from each subject at the time of vaccination (D0) and 21–28 days (D21) post-vaccination. Blood samples were processed for sera and peripheral blood mononuclear cells (PBMC). For PBMC isolation, blood was collected in CPT tubes (Becton, Dickinson and Company, Franklin Lakes, NJ, USA) at D0 and D21. These samples were processed immediately, within 1–24 hours of collection, and stored at -150˚C for future analysis. Sera was collected in SST tubes (Becton, Dickinson and Company) and processed within 24–48 hours, storing at 4˚C until separated and aliquoted for long-term storage at -30˚C. These serum samples were tested for the ability to mediate HAI and HA-specific IgG antibodies against the matching and past vaccine strains (S1 Table). Throughout the study, the H1N1 strain (A/California/7/2009) in the vaccine remained constant for three seasons, whereas the H3N2 (A/Texas/50/2012 in 2014, A/Switzerland/9715293/2013 in 2015, and A/Hong Kong/4801/2014 in 2016) vaccine strains were updated and changed each season.

**Viruses and HA antigens.**    Influenza viruses were obtained through the Influenza Reagents Resource (IRR), BEI Resources, the Centers for Disease Control and Prevention (CDC), or were provided by Sanofi Pasteur and Virapur, LLC (San Diego, CA, USA). Viruses were passaged once in the same growth conditions as they were received, in 10-day old embryonated, specific pathogen-free (SPF) chicken eggs per the protocol provided by the WHO. Titrations were performed with turkey erythrocytes and virus was standardized to 8 HAU/50 μL for use in HAI assays. A complete list of the virus strains used is provided in S1 Table.

**Recombinant HA proteins.**    Full-length HA proteins were developed for a panel of H1N1 and H3N2 IAV strains (S1 Table). Wild type and chimeric recombinant HA (rHA) proteins were expressed in EXPI293F cells and purified via a C-terminal histidine tag using HisTrap excel nickel-affinity chromatography columns (GE Healthcare Life Sciences, Marlborough, MA, USA) as previously described [8, 17, 75–77]. Purified rHA proteins were dialyzed against PBS and total protein concentration adjusted to ~1 mg/mL after BCA estimation.

**Enzyme linked immunosorbent assay (ELISA).**    Hemagglutinin-specific IgG-antibodies were quantified by ELISA as previously described [8]. Briefly, Immulon® 4HBX plates (Thermo Fisher Scientific, Waltham, MA, USA) were coated with 50 ng/well of rHA in carbonate buffer (pH 9.4) with 250 ng/mL bovine serum albumin (BSA) for ~16 h at 4˚C in humidified chambers. Plates were blocked with blocking buffer (2% BSA, 1% gelatin in PBS/0.05% Tween20) at 37˚C for 2 h. Serum samples collected from participants prior to and 21–28 days following vaccination were initially diluted 1:500 and then further serially diluted 1:2 in blocking buffer to generate 7-point binding curves. Serially diluted serum samples were added to the assay plate in duplicate and incubated ~16 h overnight at 4˚C in humidified chambers. Plates were washed 4 times with phosphate buffered saline (PBS) and HA-specific IgG detected using horseradish peroxidase (HRP)-conjugated goat anti-human IgG (Southern Biotech, Birmingham, AL, USA) at a 1:4,000 dilution and incubated for 2 h at 37˚C. Plates were then washed PBS prior to development with 100 μL of 0.1% 2,2'-azino-bis(3-ethylbenzothiazoline-6-sulphonic acid) (ABTS) solution with 0.05% $H_2O_2$ for 20 min at 37˚C. The reaction was terminated with 1% (w/v) sodium dodecyl sulfate (SDS). Colorimetric absorbance at 414nm was measured using a PowerWaveXS (Biotek, Winooski, VT, USA) plate reader. HA-specific IgG equivalent concentration was calculated based on an 8-point standard curve generated using a human IgG reference protein (Athens Research and Technology, Athens, GA, USA). Cumulative IAV HA binding was calculated by adding the IgG-equivalent of the both IAV vaccine components (H1N1 + H3N2).

**Flow cytometry.**    Human PBMC (~5 x $10^6$ live cells) were stained on ice for 30 min with fluorochrome conjugated rHA probes (180–350 pM) in 100 μL of staining buffer (PBS/2% fetal bovine serum [FBS]) as previously described [8, 30, 54, 78, 79]. Human PBMC were first treated with Fc receptor blocking solution (BioLegend, Dedham, MA, USA) then stained for 30 min on ice using titrated quantities of fluorescently conjugated monoclonal antibodies (S1 Table). After completion of surface labeling, human PBMC were washed extensively with staining buffer prior to fixation with 1.6% paraformaldehyde in staining buffer for 15 min at RT. Following fixation, cells were pelleted by centrifugation at 400x*g* for 5 min, resuspended in staining buffer and maintained at 4˚C protected from light until acquisition. Data acquisition was performed using the BD FACSARIA Fusion and analysis performed using FlowJo (FlowJo LLC, Ashland, OR, USA). Compensation values were established prior to acquisition using appropriate single stain controls. PBs were defined as CD3/CD14$^{neg}$ CD19$^+$, CD27$^+$, CD38$^{++}$ cells as previously described [67, 80].

**In vitro differentiation of B cells.**    PBMC were cultured (2 x $10^6$ viable cells/mL) in complete media containing Roswell Park Memorial Institute (RPMI) 1640 medium (Sigma, St. Loius, MO, USA) with 10% FBS (Atlanta Biologicals, Flowery Branch, GA, USA), 23.8mM

sodium bicarbonate (Fisher Scientific, Waltham, MA, USA), 7.5 mM HEPES (Amresco, Framingham, MA, USA), 170 μM Penicillin G (Tokyo Chemical Industry, Portland, OR, USA), 137 μM Streptomycin (Sigma, Burlington, MA, USA), 50 μM β-mercaptoethanol (Sigma, Burlington, MA, USA), 1 mM sodium pyruvate (Thermo Fisher Scientific, Waltham, MA, USA), essential amino acid solution (Thermo Fisher Scientific, Waltham, MA, USA), non-essential amino acid solution (Thermo Fisher Scientific, Waltham, MA, USA), 500 ng/mL R848 (Invivogen, San Diego, CA, USA) and 5 ng/mL rIL-2 (R&D, Minneapolis, MN, USA) for 7–9 days at 37˚C in 5% $CO_2$ [63, 81]. Conditioned medium supernatants were harvested and evaluated for total and rHA-specific IgG abundance by ELISA starting at a 1:5 dilution. Frequency of B cells amongst total viable PBMC was assessed by CD19 surface labeling and flow cytometry analysis.

## Statistical methods

Participants were grouped by age as previously described [25] and the response to each individual vaccine component was categorized as per the WHO and European Committee for Medicinal Products to evaluate influenza vaccines [82]. Minimal seroprotection was defined as HAI titer of 1:40 to 1:80 and participants were considered seronegative with a titer below 1:40. Statistical significance between groups was calculated using one-way ANOVA Friedman test and Dunns multiple comparisons. Values were considered significant for $p < 0.05$. Unless otherwise stated, data is presented from at least three independent experiments.

Percentage of HA binding to each vaccine strain was calculated from the cumulative IgG binding to the both IAV vaccine components for each subject individually (H1+H3). For percentage of HAI activity, serum titers were transformed to a $Log_2$ scale prior to calculation, to avoid skewness. Significant immunodominance in a group was calculated by One-sample Wilcoxon Signed rank test (%HA≠25) and 1-way ANOVA Friedman test and Dunn's multiple comparisons (H1≠H3). Statistical significance ($p < 0.05$) must be reached in both tests and the highest $p$ value is represented. Differences between pre- and post-vaccination were calculated by one-way ANOVA multiple comparisons. Percentage of rHA binding to the H3N2 vaccine component heatmap analysis was performed with GraphPad for each subject. All statistical analyses were performed using GraphPad Prism V.8.01 software.

Landscape analysis was performed on excel with average IgG antibody levels reactive to rHAs from a broad panel of current and historical IAV. Vaccine induced antibodies were calculated as the difference in rHA-reactive antibodies prior and 21–28 days post-vaccination (S10 Fig) normalized to pre-existing antibody levels as follows: $\frac{\Delta D28 - D0}{D0} X100$. Vaccine-specific HAI antibodies were determined against current and historical IAV and vaccine induced changes calculated as the fold increase in HAI prior and 21 to 28 days post-vaccination, represented in a log2 scale.

## Supporting information

**S1 Fig. Biparametric quadrant analysis of HAI titer and rHA-specific IgG (μg/mL) from 50 subjects (16 young-adult and 34 elderly) vaccinated for three consecutive years with standard of care inactivated influenza vaccine.** A-F) Profile response to the H1N1 vaccine strain. G-L) Profile response to the H3N2 vaccine strains. High-HAI antibodies in Q1, high non-HAI in Q2, strong HAI-Abs in Q3 and non-responders in Q4. Young-adult participants are depicted as red dots and elderly in blue. Doted lines represent the cohort's average for rHA-specific IgG pre-vaccination (horizontal) and the generally correlated 1:40 protective serum HAI titer (vertical). Changes in the proportion of participants in each quadrant over

time were assess by a Chi-square test ($\chi^2$).
(DOCX)

**S2 Fig.** Changes in frequency of plasmablasts of total B-cells 7 and 21–28 days after vaccination in young-adult (A) and elderly (B) participants.
(DOCX)

**S3 Fig. Frequency of rHA-reactive plasmablasts against H1N1 and H3N2 vaccine strains in young (red) and elderly (blue) subjects vaccinated over three consecutive years.**
(DOCX)

**S4 Fig. Serological antibody landscape in young-adult participants vaccinated for three consecutive years.** A-C) Serological IgG antibodies against rHA from current H1N1 vaccine strain and 4 historical seasonal H1N1 virus strains (1983–2007) in three young subjects vaccinated for three consecutive years. Colors represent antigenic similarities between H1 rHA. D-F) Serological IgG antibody levels against rHA from the current H3N2 vaccine strains and 5 historical seasonal H3N2 virus strains (1999–2011) in three young subjects vaccinated for three consecutive years. Colors represent antigenic similarities between H3 rHA.
(DOCX)

**S5 Fig. HAI antibody landscape against a broad panel of H1N1 (A-C and G-I) or H3N2 (D-F and J-L) in 6 young-adult participants vaccinated for three consecutive years.**
(DOCX)

**S6 Fig. Serological antibody landscape in young-adult participants vaccinated for three consecutive years.** A-C) Serological IgG antibodies against rHA from current H1N1 vaccine strain and 4 historical seasonal H1N1 virus strains (1983–2007) in three elderly subjects vaccinated for three consecutive years. Colors represent antigenic similarities between H1 rHA. D-F) Serological IgG antibody levels against rHA from the current H3N2 vaccine strains and 5 historical seasonal H3N2 virus strains (1999–2011) in three elderly subjects vaccinated for three consecutive years. Colors represent antigenic similarities between H3 rHA.
(DOCX)

**S7 Fig. HAI antibody landscape against a broad panel of H1N1 (A-C and G-I) or H3N2 (D-F and J-L) in 6 elderly participants vaccinated for three consecutive years.**
(DOCX)

**S8 Fig. Bmem-derived IgG antibodies against rHA from current vaccine and historical seasonal influenza virus strains in three young adult participants vaccinated for three consecutive years.**
(DOCX)

**S9 Fig. Bmem-derived IgG antibodies against rHA from current vaccine and historical seasonal influenza virus strains in three elderly participants vaccinated for three consecutive years.**
(DOCX)

**S10 Fig. Illustrative approach to calculate vaccine induced rHA-reactive antibodies every year in each analyzed participant (D#1132 in 2014 shown).** Resulting transformed data was used for panels G-L in Figs 4 and 5.
(DOCX)

**S1 Table. Key resources and reagents.**
(DOCX)

## Acknowledgments

The authors thank NIH Biodefense and Emerging Infections Research Resources Repository, NIAID, NIH for providing crucial reagents for this work. The authors would like to thank the members of the CVI protein production core, Jeffrey Ecker, Spencer Pierce, and Ethan Cooper for expression and purification of the recombinant proteins. We also thank Jonathan Murrow, Brad Phillips, Kim Schmitz, and the entire members of the UGA Clinical Trials Evaluation Unit, and give a special thanks and appreciation to the volunteer participants in the study.

## Author Contributions

**Conceptualization:** Rodrigo B. Abreu, Greg A. Kirchenbaum, Ted M. Ross.

**Data curation:** Rodrigo B. Abreu, Emily F. Clutter.

**Formal analysis:** Rodrigo B. Abreu, Giuseppe A. Sautto, Emily F. Clutter.

**Funding acquisition:** Ted M. Ross.

**Investigation:** Emily F. Clutter.

**Methodology:** Rodrigo B. Abreu, Greg A. Kirchenbaum.

**Project administration:** Rodrigo B. Abreu, Ted M. Ross.

**Resources:** Ted M. Ross.

**Supervision:** Rodrigo B. Abreu, Ted M. Ross.

**Validation:** Rodrigo B. Abreu, Emily F. Clutter.

**Visualization:** Rodrigo B. Abreu.

**Writing – original draft:** Rodrigo B. Abreu.

**Writing – review & editing:** Greg A. Kirchenbaum, Giuseppe A. Sautto, Ted M. Ross.

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
