## [Decision Letter · Decision Letter 0]

3 Mar 2021

PONE-D-20-36767

Impaired memory B-cell recall responses in the elderly following recurrent influenza vaccination

PLOS ONE

Dear Dr. Ross,

Thank you for submitting your manuscript to PLOS ONE. After careful consideration, we feel that it has merit but does not fully meet PLOS ONE’s publication criteria as it currently stands. Therefore, we invite you to submit a revised version of the manuscript that addresses the points raised during the review process.

While the reviewers found both the project and the goals that the authors are trying to achieve to be of significant interest, there were some substantial issues with the data presentation and interpretation of the findings that were identified by the reviewers.  Specifically, during the revision process the authors are asked to make certain that the data support the conclusions that were drawn.  Also, please address the study design and technical aspects that greatly affect data presentation and interpretation of the findings.

We look forward to receiving your revised manuscript.

Kind regards,

Victor C Huber

Academic Editor

PLOS ONE

Journal Requirements:

2. Please provide additional details regarding participant consent. In the ethics statement in the Methods and online submission information, please ensure that you have specified whether consent was informed.

3. In your Methods section, please provide additional information about the participant recruitment method and the demographic details of your participants. Please ensure you have provided sufficient details to replicate the analyses such as:

a) a table of relevant demographic details,

b) a statement as to whether your sample can be considered representative of a larger population, and

c) a description of how participants were recruited.

4.Our staff editors have determined that your manuscript is likely within the scope of our Call for Papers on Influenza. This editorial initiative is headed by PLOS ONE Guest Editors Dr. Meagan Deming and Dr. Deshayne Fell. The Collection encompasses research on influenza prevention on every level, including in vitro, translational, behavioral, and clinical studies; disease and immunity modelling; as well as new approaches to influenza prevention. Additional information can be found on our announcement page: https://collections.plos.org/call-for-papers/influenza/.

Currently, your manuscript is included in the group of papers being considered for this call. Please note that being considered for the Collection does not require additional peer review beyond the journal’s standard process and will not delay the publication of your manuscript if it is accepted by PLOS ONE. We would greatly appreciate your confirmation that you would like your manuscript to be considered for this Collection by indicating this in your next cover letter. If you would prefer to remove your manuscript from collection consideration, please specify this in your cover letter.

5.We note that the grant information you provided in the ‘Funding Information’ and ‘Financial Disclosure’ sections do not match.

8. Thank you for stating the following in the Financial Disclosure section:

"TMR:  Received Award; HHSN272201400004C (NIAID Centers of Excellence for Influenza Research and Surveillance, CEIRS).  National Institutes of Health; www.nih.gov; The sponsors did not play any role in the  in the study design, data collection and analysis, decision to publish, or preparation of the manuscript."

We note that one or more of the authors are employed by a commercial company: Cellular Technology Limited

9.We noticed you have some minor occurrence of overlapping text with the following previous publication, which needs to be addressed:

-https://www.frontiersin.org/articles/10.3389/fimmu.2020.00902/full (Discussion, paragraph 2)

In your revision ensure you cite all your sources (including your own works), and quote or rephrase any duplicated text outside the methods section. Further consideration is dependent on these concerns being addressed.

Reviewers' comments:

Reviewer's Responses to Questions

**Comments to the Author**

1. Is the manuscript technically sound, and do the data support the conclusions?

Reviewer #1: Yes

Reviewer #2: Partly

Reviewer #3: Partly

2. Has the statistical analysis been performed appropriately and rigorously? 

Reviewer #1: Yes

Reviewer #2: No

Reviewer #3: I Don't Know

3. Have the authors made all data underlying the findings in their manuscript fully available?

Reviewer #1: Yes

Reviewer #2: No

Reviewer #3: Yes

4. Is the manuscript presented in an intelligible fashion and written in standard English?

Reviewer #1: Yes

Reviewer #2: Yes

Reviewer #3: Yes

5. Review Comments to the Author

Reviewer #1: PONE-D-20-36767

Impaired memory B-cell recall responses in the elderly following recurrent influenza vaccination

Abreu et al

In this manuscript the authors track serological and memory B cell responses to influenza vaccination in young and older adults, over three seasons with repeated vaccination. HAI responses in elderly were transient and narrow in breadth compared to those in young adults, and this correlated with a reduced peripheral plasmablast response.

The topic is of great interest and this approach is likely to lead to important contributions to the field. The manuscript is well written and clear and the techniques are well chosen and properly controlled. Statistical analysis seems to be appropriately performed. I noticed a handful of typos:

• p. 11, “has been were updated 7 times”

• p. 14, “The here reported antigenic competition” (should probably be “the antigenic competition reported here”)

• p. 5, 15, “sera was” (should be “were”)

• Fig 1, “H3N12-reative”

• Fig S3, “in you and elderly subjects”

Some references contained extraneous text and links, e.g.

• Reference 9, “REFERENCES Linked references are available on JSTOR for this article : You may need.”

• Ref 37, https://europepmc.org/backend/ptpmcrender.fcgi?accid=PMC2065290&blobtype=pdf.

• Ref 39, “http://www.ncbi.nlm.nih.gov/pubmed/25590680%0Ahttp://www.pubmedcentral.nih.gov/articlerender.fcgi?artid=PMC4584793.”

My main concern with this paper is that in some cases the data may be overinterpreted. The main message I see in the data is that there is extreme individual-to-individual, and year-to-year, variation, and finding trends in this variation is very difficult. The authors acknowledge this (e.g. “we observe tremendously different responses to influenza virus vaccination”, p. 7), and this may be the most important message from this and similar studies; trying to find patterns where there are none is misleading.

p. 5, “In young adult participants, recurrent vaccination with the exact same vaccine strain (i.e. H1N1) induced long-term persistent changes in the serological profile towards receptor-binding epitopes …” doesn’t reference a figure – Where is this shown? I don't see this convincingly in Figure 1

p. 5, “participants were categorized as high-HAI (Q1), high-non-HAI (Q2), strong-HAI (Q3) serological profiles” – In Figure S1 the values don’t seem to naturally separate into high/low groups but rather look more like a (log)- normal distribution. I understand the point of splitting the HAI at the standard CoP value of 40, but the non-HAI division at 100 looks arbitrary, and it seems that a very small change (splitting at 90 instead of 100?) might have changed the interpretation significantly. Is there an objective reason for drawing the distinction there?

p. 6, “Young adult participants had a prominent increase in B-cell PBs every year following vaccination (Fig 3B and S2)” – This doesn’t seem to be true for all subjects, and looks more like individual variation than a trend to me. Is there statistical support for this statement?

Do we know previous vaccine/infection history? Were any of the subjects vaccinated in 2013 (making 2014 a repeat season)? Were e.g., young but not old subjects infected with H1N1pdm09 during the pandemic?

Reviewer #2: In their paper, Abreu and colleagues compare antibody and B cell responses to influenza in elderly vs. younger individuals over a three-year period. Strengths of their study include the importance of the study topic, the demonstration of alterations in serologic and B cell responses to vaccine vs. historical strains and the long period of follow-up. Weaknesses, include overstatements of what their data show with respect to antibody reactivity, awkward figure layouts with unclear figure legends and potential technical issues with the immunophenotyping.

Major concerns:

Cross-reactivity cannot be directly demonstrated without studying individual antibodies. Hence statements such as the one on p. 11 of the combined manuscript file in the results section, “In parallel, recurrent vaccination with antigenically distinct strains resulted in ~45% of young participants acquiring cross-reactive HAI activity to the new H3N2 strain in 2015 prior to vaccination,” are problematic. I would recommend rephrasing this text to make it clear that you are not claiming that these antibodies are “cross-reactive.”

There are also very broad statements about adaptation of the antibody response to “drifted epitopes” (e.g., on page 18 of the merged pdf file where the discussion section references Fig. 1). As no data directly testing specificity of antibodies to specific epitopes were presented in Fig. 1, so I would remove these claims.

Many of the data for the comparative studies of antibody levels, referenced to IgG, are not presented in a sufficiently detailed manner to evaluate the adequacy of the methods for the claims that are being made. Based on the methods, it looks like the authors are generating titration curves, but it is not clear from the figures how these titrations are being used to create ratio values.

The immunophenotyping analysis has potential technical issues with respect to the specificity of probe staining and gating. It looks like only a single fluorophore was used to identify each of the antigen-enriched cell populations (one fluorophore per probe), but it is well known that there can be a high level of background with these assays, necessitating approaches where each probe is separately labeled with at least two different fluorophores. This is potentially a major issue that impacts conclusions about historical reactive Bmem cells being higher than vaccine reactive, for example. I would recommend re-testing some samples with double labeled antigen probes to make sure that this result holds up with a cleaner flow cytometric analysis. Otherwise it could be due to something trivial, like the historical HA probe has more noise on one of the fluorophores than the vaccine probe on a different fluorophore etc..

Along similar lines, some of the “broadly reactive” cells could be noise. Were dead cells and doublets rigorously gated out? Some of these events may be noise and many of the others in this “double positive” gate may well be single positive because the dots in some of the plots are very near the vaccine gate, raising the issue of how the gates are defined. Do you have any controls that you can use to justify the position of the gates etc.? Ultimately, proving that these “double positive” B cells actually are broadly reactive would require more definitive experiments such as cloning mAbs and demonstrating their individual binding reactivities. I would recommend making this caveat in the discussion.

Minor concerns and other comments:

In Fig. 1A it looks like the legend for older and younger subjects is reversed from the rest of the figure (and the rest of the paper for that matter). Is this an error?

The antibody classifications used for the pie charts in Fig. 1B seem arbitrary and not as helpful as showing the dot plots in the supplementary figure as many of the subjects appear to fall on the boundaries between the different categories which are not really dichotomous variables but continuous ones. I would suggest putting the dot plots from the supplementary figure into figure 1 (which are actually quite clear, unlike the pie charts)…

Which time point is being shown in Fig. 1B? The legend references the methods, but the methods doesn’t reveal this, unless I am missing something.

For Fig. 2, binding appears to be skewed towards H1 but HAI is skewed towards H3 which is interesting. Could non-HA specificities be contributing to virus neutralization?

Do the grey boxes in the heatmaps indicate an intermediate value or no data? Please indicate this in the figure legend.

Fig. 3 panel A, please indicate what is meant by vaccine vs. historical HA. For example, if you have an individual who received a vaccine in 2016, what would be considered historical (2015+2014 or just 2015 etc.)?

Fig. 3 panel B, what do you make of several individuals who appear to have higher PB frequencies on D0 in 2016? Were all of these samples processed on the same day or run on the same day by FACS?

In the text where you comment on plasmablast expansions following vaccination (or the lack thereof in the elderly), I would reference other literature that also documents this.

For panel 3D and E, it would be more convincing to show a full time course starting with D0, not just on D7…. (If you have the data, why not show a similar plot in Fig. 3 to the one with memory B cells in Fig. 6?)

Fig. S3 legend suggests that there are data points for young (typo?) and elderly, why not color code the dots by young vs. elderly?

Fig. S4-S9, it is hard to see changes in antibody levels. Why not show all of these data as a heat maps?

Fig. S10 seems incomplete. Why not include the D7 time point also and make two separate figures for young vs. old?

For subset analysis, absolute counts may matter more than the relative fractions. I realize it may be hard to get these, but if you have them, I would recommend including the data.

ABCs are more commonly referred to as age-associated B cells than atypical B cells. Some ABCs can be CD27+ so using the term “double negative (CD27-, IgD-) ABCs” is confusing. If you used other markers such as CD21, CXCR5 or CD11c to gate specifically on ABCs, please indicate this in the methods section. If you didn’t use any other ABC markers, I would refer to these cells as DN (IgD-, CD27-) and not as ABCs.

Reviewer #3: The purpose of this study is to establish how flu-specific memory B cells and serum antibody are shaped by recurrent vaccinations in young versus elderly populations. They follow young adult and elderly populations who received the inactivated vaccine 3 years in a row (2014-15 through 2016-17). Notably, the vaccine H1 strain remained constant, while the H3 changed each year).

While some interesting observations are made (e.g. longitudinal changes in total HA and HAI titers), several conclusions are not sufficiently supported by the data due to several concerns. These include small subject numbers and significant variation within subject groups ; unusual or unclear statistical methods, concern regarding background signal for the fluorochrome-labeled HA probes used in the FACS analyses. Finally, an overarching problem is whether the observed vaccine-generated responses indeed represent recall vs. novel clones, and this is not addressed in the paper. Clonal analyses would be a superior method to answer many of their questions.

Specific comments:

Figure 1:

-I'm not sure about the validity of their methodology for joint titer/HAI analysis, and I think the grouping of subjects into these quadrants and presenting summary data as pie charts makes it difficult to interpret the longitudinal changes. They claim to show persistent serological changes in young and transient changes in elderly, but I'm not seeing this in the data as presented. Showing the analysis in a single dimension (with each individual as a point) might be more useful (e.g. HAI titer before and after vaccination, rHA titer before and after). As the data is currently shown, they should be more clear about their statistical analysis.

-Vaccination seems to boost titers and HAI in all instances except for H1 in the young with 2015 vaccine - why?

-Panels 1B and 1C aren't mentioned in the results section text

-They claim to show generation of cross-reactive H3 responses, but this cannot be concluded from the data as shown (unless I'm missing something). There is no way to differentiate recall vs. novel responses in these analyses.

Figure 2:

-We are shown fractions of total HA-specific titers, but how much H1 and H3 Ab (mg/ml) is there? Different frequencies in old vs. young could still represent equivalent titers, so absolute titers may be more useful (or should at least be included so this "balance" data can be interpreted better)

Figure 3:

-Plasmablast frequencies are expected to be "messy" since these represent a tiny fraction of total B cells, but as such, the conclusions made herein with a small number of subjects are a stretch. For example, it is claimed young have a "prominent" PB expansion following vaccination whereas the elderly do not, but many young don't appear to make a detectable PB response either. As far as I can tell, there was no quantitative comparison between young vs. old.

-Arrows are missing from the gating scheme in 3A, so one must surmise what gates were chosen for the downstream plots.

-HA probe staining is a concern. Most analyses of this type have employed probes that are separately labelled with two distinct fluorochromes so that only events on the diagonal (i.e.; positive for both probes) are counted, and the events on the x- or y- dimensions are considered non-specific noise. THe frequencies of HA binding cells in some of their plots are rather high. It would be best if these were rpeated with dual probes, and the inclusion of some type of control to demonstrate the positive signal is real. Perhaps show PBs from an individual far from vaccination? Or at the pre-vaccination time point? Or PBs from a young (flu-naive) child?

-I'm confused what the frequencies in 3D and 3E represent

-Fig S2 is not evidence of a highly-specific (vs. broadly reactive) vaccine response. Is this a typo?

Figure 6:

-Again, a control for the HA staining would be useful - the frequency of HA-binding for the vaccine strain are high

-"However, when tracking the dynamics of HA-specific CS-Bmems over time, there was an increase in vaccine-specific CS-Bmems at the cost of reactive cells to historical strains 7-9 days after vaccination (Fig. 6C-D)." Percentages are being shown here, so the increase in vaccine-specific cells should not be described in a way that suggests historical strain-reactive cells are being lost, since absolute numbers might be the same but other pools may have increased.

-It should be noted (6I) that the B cell compartment in young vs. old individuals is quite different in the proportional representation of lymphocyte subsets, and that this difference could impact ASC differentiation of in vitro stimulated lymphocytes. This is a significant caveat to the experiment.

-What does "IgG conditioned" mean? This is not made clear from the text.

Figure 7:

-HA staining again looks suspect - over 7% of ABCs binding the vaccine probe – this is rather high (unless there is an aspect of pre-gating not evident from the figure.

6. PLOS authors have the option to publish the peer review history of their article (what does this mean?). If published, this will include your full peer review and any attached files.

Reviewer #1: No

Reviewer #2: No

Reviewer #3: No

---

## [Author Response · Author response to Decision Letter 0]

13 May 2021

Rebuttal letter PONE-D-20-36767

Impaired memory B-cell recall responses in the elderly following recurrent influenza vaccination

Abreu et al.

Reviewer 1:

The authors appreciate the reviewer’s view on the topic and relevance of the study. We apologize for some typographical and reference errors throughout the manuscript, these have now been corrected. 

Regarding the major scientifically concerns please see below our perspective on some topics and how we addressed them in the main manuscript. 

1. My main concern with this paper is that in some cases the data may be overinterpreted. The main message I see in the data is that there is extreme individual-to-individual, and year-to-year, variation, and finding trends in this variation is very difficult. The authors acknowledge this (e.g. “we observe tremendously different responses to influenza virus vaccination”, p. 7), and this may be the most important message from this and similar studies; trying to find patterns where there are none is misleading.

Authors: We recognized that with only 12 individuals, it is impossible to see statistically significant trends, and individual variation tends to stand-out. However, volunteer recruitment for longitudinal studies such as this one is challenging, and the amount of sample required for deep immunoprofiling through different immunological assays greatly decreases the number of subjects available for the study. Nonetheless, this and other published studies (sometimes based on 1, 2 or 3 individuals) seem to point towards a common trend. Despite all variations, young vaccinees seem to develop and adapt their repertoire to newly drifted strains, while elderly vaccinees, recall pre-existing non-neutralizing antibodies. We do not intend to mislead into false conclusions, but merely describe our observations and offer our biological interpretation. Perhaps in the near future, metadata analysis of multiple different studies with small samples sizes will provide robust statistical evidence for the conclusions here presented. 

2. p. 5, “In young adult participants, recurrent vaccination with the exact same vaccine strain (i.e. H1N1) induced longterm persistent changes in the serological profile towards receptor-binding epitopes …” doesn’t reference a figure – Where is this shown? I don't see this convincingly in Figure 1

Authors: Figure 1 and S1 shows a biparametric (HAI by IgG) analysis of 16 young-adult and 34 elderly subjects’ serological responses to H1N1 and H3N2 vaccine components over three consecutive years. Looking at the serological profile to the H1N1 vaccine pre-vaccination in 2015 and 2016 (year 2 and 3), we observe that the vast majority of young vaccinees have low HA-specific antibody levels with high HAI activity, while elderly vaccinees regress to a profile similar to 2014 pre-vaccination with high HA-specific antibody levels with low-HAI activity. The same does not seem to hold true when the vaccine strain is consecutively updated. For the H3N2 vaccine strain, both young and elderly vaccinees show a transient rise in HAI+ antibodies but these either fail to neutralize the updated strain, or do not persist in circulation for the subsequent year. 

3. p. 5, “participants were categorized as high-HAI (Q1), high-non-HAI (Q2), strong-HAI (Q3) serological profiles” – In Figure S1 the values don’t seem to naturally separate into high/low groups but rather look more like a (log)- normal distribution. I understand the point of splitting the HAI at the standard CoP value of 40, but the non-HAI division at 100 looks arbitrary, and it seems that a very small change (splitting at 90 instead of 100?) might have changed the interpretation significantly. Is there an objective reason for drawing the distinction there?

Authors: We understand the reviewer’s comment regarding the cut-off value for high and low HA- specific IgG antibody levels. Despite seeming arbitrary, this is actually based on the cohort average value pre-vaccination (this is now clearly stated in the supplementary figure caption). We tested for the impact of using a different cut-off value, such as the 1st or 3rd quartile. Some subjects would move across quadrants but the overall conclusion is that young vaccinee subjects mainly retained H1N1 HAI+ antibodies.

4. p. 6, “Young adult participants had a prominent increase in B-cell PBs every year following vaccination (Fig 3B and S2)” – This doesn’t seem to be true for all subjects, and looks more like individual variation than a trend to me. Is there statistical support for this statement?

Authors: Unfortunately, due to the small sample size we cannot back-up this conclusion with a statistical value. What we observe is that out of the 6 tested young adults, 4 showed an increase in B-cell PBs every year following vaccination. In contrast, only one elderly vaccinee (D#1089) showed a small increase in B-cell PBs in two consecutive years post vaccination (only in 2015 and 2016). As mentioned before we agree that a larger sample size would be extremely beneficial, but the number of volunteers available is extremely reduced for such studies. 

5. Do we know previous vaccine/infection history? Were any of the subjects vaccinated in 2013 (making 2014 a repeat season)? Were e.g., young but not old subjects infected with H1N1pdm09 during the pandemic?

Authors: We understand the reviewer’s question and recognize that recent influenza infection or vaccination can impact the response to influenza vaccination, but unfortunately, we do not have access to clinical or vaccination history of these participants prior to the study enrolment. Future studies will try to address the impact of influenza vaccination over the last 13 years (before the H1N1pdm09 outbreak). Even if not based on longitudinal samples, it should clarify the impact of infection with H1N1pdm09 in the response to IAV vaccination. 

 

Reviewer 2:

The authors appreciate the reviewer’s constructive criticism. Please see below a point-by-point reply to the concerns presented. 

Major concerns:

1. Cross-reactivity cannot be directly demonstrated without studying individual antibodies. Hence statements such as the one on p. 11 of the combined manuscript file in the results section, “In parallel, recurrent vaccination with antigenically distinct strains resulted in ~45% of young participants acquiring cross-reactive HAI activity to the new H3N2 strain in 2015 prior to vaccination,” are problematic. I would recommend rephrasing this text to make it clear that you are not claiming that these antibodies are “cross-reactive.” 

Authors: We understand the reviewer’s comment regarding the limitations of polyclonal studies. We recognize that monoclonal identification and characterization can show the development of truly broadly-reactive antibodies, however it generally does not assess the seroprevalence of such clones and their relevance in the overall serological response to vaccination. Here, we do not claim that the vaccine elicits broadly-reactive antibodies, but rather that the vaccine elicits a cross-reactive antibody profile. Regarding the specific comment on p11 we added serological cross-reactive HAI activity to clarify the reader that we did not assess specific monoclonals. 

2. There are also very broad statements about adaptation of the antibody response to “drifted epitopes” (e.g., on page 18 of the merged pdf file where the discussion section references Fig. 1). As no data directly testing specificity of antibodies to specific epitopes were presented in Fig. 1, so I would remove these claims.

Authors: Obviously epitope mapping is not possible with a polyclonal mixture or it would need very sophisticated technologies (e.g., cryo-EM) not generally available. Nonetheless, the emergence of HAI activity against drifted strains, in absence of significant increases in overall HA-specific IgG levels, is indicative of changes/adaptation of the antibody reactivity profile. 

Many of the data for the comparative studies of antibody levels, referenced to IgG, are not presented in a sufficiently detailed manner to evaluate the adequacy of the methods for the claims that are being made. Based on the methods, it looks like the authors are generating titration curves, but it is not clear from the figures how these titrations are being used to create ratio values.

Authors: rHA-reactive antibody levels were measured as an absolute IgG equivalent value by ELISA based on an IgG standard curve as previously described (Sautto et al., 2018. ImmunoHorizons; Abreu et al., 2020. JCI Insight; Abreu et al., 2020. Front. Imm.; Forgacs et al., PlosOne. 2021). Each sample was run in triplicate in a 7-point dilution curves, and points that fell within the standard curve range were averaged for a final estimation of the absolute rHA-reactive IgG equivalent in the serum. This method allows for robust quantification of antigen-specific antibodies with minimal inter-day assay variability, ideal for comparison of longitudinal samples that need to be measured across different days.

The immunophenotyping analysis has potential technical issues with respect to the specificity of probe staining and gating. It looks like only a single fluorophore was used to identify each of the antigen-enriched cell populations (one fluorophore per probe), but it is well known that there can be a high level of background with these assays, necessitating approaches where each probe is separately labeled with at least two different fluorophores. This is potentially a major issue that impacts conclusions about historical reactive Bmem cells being higher than vaccine reactive, for example. I would recommend re-testing some samples with double labeled antigen probes to make sure that this result holds up with a cleaner flow cytometric analysis. Otherwise it could be due to something trivial, like the historical HA probe has more noise on one of the fluorophores than the vaccine probe on a different fluorophore etc..

Along similar lines, some of the “broadly reactive” cells could be noise. Were dead cells and doublets rigorously gated out? Some of these events may be noise and many of the others in this “double positive” gate may well be single positive because the dots in some of the plots are very near the vaccine gate, raising the issue of how the gates are defined. Do you have any controls that you can use to justify the position of the gates etc.? 

Authors: The reviewer’s concern regarding antigen specific B-cell quantification through probe staining is legitimate, however the use of these probes has been extensively reported in the literature by our and other groups. The use of two channels for each rHA (4 channels in total) is not compatible with our flow-cytometry B-cell panel. Specific probe staining was validated with HA-specific hybridomas and with reference donor samples (Ecker et al., 2021. Vaccines; Abreu et al., 2020. JCI Insight), where frequencies of H1- and H3- (present and historical) specific Bmems were measured in independent experiments with interchangeable fluorochrome labeling and with consistent results. Furthermore, these reagents (made in house) have been used and validated extensively in collaborative work across the CEIRS and CIVIC network (manuscripts in preparation). In particular, a successful antibody repertoire sequencing of H1, H3 and H1/H3 HA-probe sorted Bmems has been performed, confirming the HA specificity of the corresponding expressed mAbs.

Ultimately, proving that these “double positive” B cells actually are broadly reactive would require more definitive experiments such as cloning mAbs and demonstrating their individual binding reactivities. I would recommend making this caveat in the discussion.

Authors: We agree with the reviewer’s comment on the need for expression and purification of these broadly-reactive antibodies. As previously mentioned this work has been performed and the manuscript is in preparation. Despite understanding the value of this work, we truly believe on the importance and need to better understand the changes on the complex polyclonal response following influenza vaccination. In our opinion, it is important to recognize that the serological response is much more than simply the additive effect of some monoclonal antibodies. 

Minor concerns and other comments: In Fig. 

1A it looks like the legend for older and younger subjects is reversed from the rest of the figure (and the rest of the paper for that matter). Is this an error?

Authors: We sincerely apologize for the oversight and appreciate the reviewer’s attention. This has been corrected. 

The antibody classifications used for the pie charts in Fig. 1B seem arbitrary and not as helpful as showing the dot plots in the supplementary figure as many of the subjects appear to fall on the boundaries between the different categories which are not really dichotomous variables but continuous ones. I would suggest putting the dot plots from the supplementary figure into figure 1 (which are actually quite clear, unlike the pie charts)…

Which time point is being shown in Fig. 1B? The legend references the methods, but the methods doesn’t reveal this, unless I am missing something.

Authors: We apologize for the oversight, timepoints are now clearly stated on the figure legend. Serum samples were collected prior to and 21-28 days post-vaccination over three consecutive years (2014-2016). The authors understand that the dot plots might be easier to follow by those used to multidimensional data, however each reviewer had a different opinion regarding this figure. Therefore, we decided to keep figure 1 as originally submitted. 

For Fig. 2, binding appears to be skewed towards H1 but HAI is skewed towards H3 which is interesting. Could non- HA specificities be contributing to virus neutralization? Do the grey boxes in the heatmaps indicate an intermediate value or no data? Please indicate this in the figure legend. 

Authors: The authors agree that this is an interesting phenomenon. We have previously reported this (Abreu et al., 2020. JCI Insight; Nuñez et al., 2017. Plos One) in 18-65 y.o. vaccinees, but it now seems to be exacerbated in elderly participants. In our opinion, this might be driven by early-life influenza exposure (imprinting), the continuous update of certain vaccine strains (H3) in presence of other constant antigens (H1 and IBV) and intrinsic immunogenicity of the HA. Regarding the possibility of non-HA specific neutralizing antibodies (e.g., NA-directed) we did not see a measurable rise in NA-antibodies following vaccination, and up to 2016 influenza vaccine manufacturers did not disclose NA content in their vaccines. Additionally, this is consistent with what was already described in the literature where NA-directed antibodies are mainly elicited following influenza infection and not vaccination (Chen et al., 2018. Cell). Future studies will also continue to elucidate the main driving mechanism for such pronounced immunodominance of certain vaccine components. Grey boxes represent missing values. This is now clearly stated in the figure legend

Fig. 3 panel A, please indicate what is meant by vaccine vs. historical HA. For example, if you have an individual who received a vaccine in 2016, what would be considered historical (2015+2014 or just 2015 etc.)? 

Authors: This information is now clearly stated in the figure legend. H1N1 Vac rHA is CA/09 for 2014-16, and H1N1 Hist. rHA are NC/99 and Sing/86 (pooled at half concentration); H3N2 Vac rHA is TX/12 for 2014, Switz/13 for 2015 and HK/14 for 2016; H3N2 Hist rHA are Pan/99 and Wisc/05 for 2014-16.

Fig. 3 panel B, what do you make of several individuals who appear to have higher PB frequencies on D0 in 2016? 

Were all of these samples processed on the same day or run on the same day by FACS? In the text where you comment on plasmablast expansions following vaccination (or the lack thereof in the elderly), I would reference other literature that also documents this.

Authors: Samples were processed and analyzed in two separate experiments of 3 young and 3 elderly participants. All samples from the same donor were processed and analyzed concurrently. In our opinion the elevated PB frequencies in some elderly participants (1089, 1132 and 1051) is likely the result of some inflammatory pre-condition. In fact, 1089 shows consistent high PB frequencies perhaps indicative of a chronic inflammatory disease. Common age-associated diseases were not recorded, and their presence was not an exclusion criteria for this study. This possibility is discussed (line 315 now highlighted) and these limitation clearly presented. We understand it is speculative but in the absence of clinical records it is the best explanation we can provide. 

For panel 3D and E, it would be more convincing to show a full time course starting with D0, not just on D7…. (If you have the data, why not show a similar plot in Fig. 3 to the one with memory B cells in Fig. 6?) 

Authors: Figure 3 refers to the frequency of plasmablast in the periphery, which transiently rises 7-9 days after vaccination. In steady state or later stages of the immune response, the frequency of plasmablasts in circulation is very low. Therefore, looking at the frequency of antigen specific cells in such limited number of events would lead to aberrant results. Of the three elderly subjects with aberrant high plasmablast frequencies, we could detect a small frequency of Vac-specific plasmablasts (<5% in D#1089) pre-vaccination in 2014 and 2016, but undetectable 28 days post-vaccination. At this point we do not have a reasonable explanation for this observation. Future studies will focus on donors with a similar profile to see if this has any biological meaning. 

Fig. S3 legend suggests that there are data points for young (typo?) and elderly, why not color code the dots by young vs. elderly?

Authors: We have colored young and elderly subjects as suggested.

Fig. S4-S9, it is hard to see changes in antibody levels. Why not show all of these data as a heat maps?

Authors: We have previously published similar data from the entire cohort as heatmaps that clearly showed the impact of age on the response to influenza vaccination (Nuñez et al., 2017. Plos One). Based on the feedback from that manuscript it does not seem like a heatmap properly conveys the landscape of each donor. In our opinion, and based on previous literature (Fonville et al., 2013, Science) this approach better depicts magnitude changes in the reactivity to influenza strains from a specific era. 

Fig. S10 seems incomplete. Why not include the D7 time point also and make two separate figures for young vs.old?

Authors: Fig S10 refers to changes in the serological antibody landscape following influenza vaccination. It is merely a visual representation of data transformation that feed into panels G-L of figure 4 and 5. Serological responses were not assessed 7 days after vaccination. 

For subset analysis, absolute counts may matter more than the relative fractions. I realize it may be hard to get these, but if you have them, I would recommend including the data.

Authors: We agree with the reviewer’s comment on the value of absolute counts. Unfortunately, we did not run absolute counting beads. 

ABCs are more commonly referred to as age-associated B cells than atypical B cells. Some ABCs can be CD27+ so using the term “double negative (CD27-, IgD-) ABCs” is confusing. If you used other markers such as CD21, CXCR5 or CD11c to gate specifically on ABCs, please indicate this in the methods section. If you didn’t use any other ABC markers, I would refer to these cells as DN (IgD-, CD27-) and not as ABCs.

Authors: We recognize that ABC nomenclature, function and phenotype is not yet standardized. Some groups refer to age-associated B-cells, while in infectious diseases (malaria and HIV) these are called atypical B-cells. In this study we could not assess CD21, CXCR5, CD11c or T-bet expression, but in future studies we will focus on the transcriptional profile of this population in high and low vaccine responders. We have changed every reference to ABC in the text for DN-C (double negative B-cells).

 

Reviewer #3: 

The purpose of this study is to establish how flu-specific memory B cells and serum antibody are

shaped by recurrent vaccinations in young versus elderly populations. They follow young adult and elderly populations who received the inactivated vaccine 3 years in a row (2014-15 through 2016-17). Notably, the vaccine H1 strain remained constant, while the H3 changed each year).

While some interesting observations are made (e.g. longitudinal changes in total HA and HAI titers), several conclusions are not sufficiently supported by the data due to several concerns. These include small subject numbers and significant variation within subject groups ; unusual or unclear statistical methods, concern regarding background signal for the fluorochrome-labeled HA probes used in the FACS analyses. Finally, an overarching problem is whether the observed vaccine-generated responses indeed represent recall vs. novel clones, and this is not addressed in the paper. Clonal analyses would be a superior method to answer many of their questions.

The authors appreciate the interest shown in this work and the reviewer’s comments. Please see bellow a point-by-point reply to the concerns presented. 

Specific comments:

Figure 1:

-I'm not sure about the validity of their methodology for joint titer/HAI analysis, and I think the grouping of subjects into these quadrants and presenting summary data as pie charts makes it difficult to interpret the longitudinal changes.

Authors: HAI titer and rHA-specific antibodies were measured in matching samples collected prior and post vaccination from participants vaccinated over three consecutive years. Both parameters should be related and can expose different polyclonal profiles, such as strong responses towards the receptor-biding site. 

They claim to show persistent serological changes in young and transient changes in elderly, but I'm not seeing this in the data as presented. Showing the analysis in a single dimension (with each individual as a point) might be more useful (e.g. HAI titer before and after vaccination, rHA titer before and after). As the data is currently shown, they should be more clear about their statistical analysis

Authors: The serological profile of young-adults against the H1N1 vaccine strain pre-vaccination in 2015 and 2016 is reflective of the changes in 2014 after vaccination. Changes in the proportion of participants in each quadrant over time were assess by a Chi-square test (�2) and this is now clearly stated in the figure legend.

-Vaccination seems to boost titers and HAI in all instances except for H1 in the young with 2015 vaccine. Why?

Authors: In 2015 young adults had skewed antibody responses towards the H3N2 vaccine component possibly due to differences in strain immunogenicity and early-life imprinting. Besides, young adults retain strong HAI antibody titers even one year after vaccination which could contribute to a phenomenon generally known as “antigen trapping” (Lesser et al. 2012, PLOS Path; Kim et al. 2009, J. Immunology; Miller et al. 2013, Sci. Transl. Med.).

-Panels 1B and 1C aren't mentioned in the results section text.

Authors: Figure 1B and 1C in-text callouts were added. 

-They claim to show generation of cross-reactive H3 responses, but this cannot be concluded from the data as shown (unless I'm missing something). There is no way to differentiate recall vs. novel responses in these analyses.

Authors: We agree that a polyclonal approach cannot prove the rise of cross-reactive monoclonal antibodies, but the landscape profile definitely shows cross-reactive serological polyclonal responses following influenza vaccination. Indeed, we cannot clarify at this point how much of the response derives from pre-existing immunity clonotype expansion or de novo B-cell memory, which can only be shown through a sequence-based approaches. Subsequent studies will focus on clonotype expansion in young and elderly participant following recurrent influenza vaccination. 

Figure 2:

-We are shown fractions of total HA-specific titers, but how much H1 and H3 Ab (mg/ml) is there? Different frequencies in old vs. young could still represent equivalent titers, so absolute titers may be more useful (or should at least be included so this "balance" data can be interpreted better).

Authors: Antibody levels against H1 and H3 were highly variable across participants ranging from 1-1000 ug/mL. Assuming the reviewer means end-point titers, which is generally a Log2 assay, absolute IgG equivalent units provide a much more accurate and robust estimate of serological antibody levels. From a data analysis perspective, to compare the reactivity across vaccine strains a continuous variable is much more valuable than a discrete one like what we would obtain from endpoint titer. 

Figure 3:

Plasmablast frequencies are expected to be "messy" since these represent a tiny fraction of total B cells, but as such, the conclusions made herein with a small number of subjects are a stretch. For example, it is claimed young have a "prominent" PB expansion following vaccination whereas the elderly do not, but many young don't appear to make a detectable PB response either. As far as I can tell, there was no quantitative comparison between young vs. old.

Authors: We recognize that a major limitation of this study is the small sample size. With only 12 individuals it is impossible to see statistically significant trends, and individual variation tends to stand out. However, volunteer recruitment for longitudinal studies such as this one is challenging, and the amount of sample required for deep immunoprofiling through different immunological assays greatly decreases the number of subjects available for the study. Nonetheless, this and other studies (sometimes based on 1, 2 or 3 individuals) seem to point towards a common trend. Despite all variation, 4 out of the 6 young adults tested showed an increase in B-cell PBs every year following vaccination. In contrast, only one elderly vaccinee (D#1089) showed a small increase in B-cell PBs in two consecutive years post vaccination (only in 2015 and 2016).

-Arrows are missing from the gating scheme in 3A, so one must surmise what gates were chosen for the

downstream plots.

Authors: The authors do not understand the reviewer’s comment. Figure 3A has arrow tracking the gating strategy. 

-HA probe staining is a concern. Most analyses of this type have employed probes that are separately labelled with two distinct fluorochromes so that only events on the diagonal (i.e.; positive for both probes) are counted, and the events on the x- or y- dimensions are considered non-specific noise. THe frequencies of HA binding cells in some of their plots are rather high. It would be best if these were rpeated with dual probes, and the inclusion of some type of control to demonstrate the positive signal is real. Perhaps show PBs from an individual far from vaccination? Or at the pre-vaccination time point? Or PBs from a young (flu-naive) child?

Authors: The reviewer’s concern regarding antigen specific B-cell quantification through probe staining is legitimate, however the use of these probes has been extensively reported in the literature (by our and other groups). The use of two channels for each rHA (4 channels in total) is not compatible with our flow-cytometry B-cell panel. Specific probe staining was validated with HA-specific hybridomas and with reference donor samples (Ecker et al., 2021. Vaccines; Abreu et al., 2020. JCI Insight), where frequencies of H1 and H3 (present and historical) specific Bmems were measured in independent experiments with interchangeable fluorochrome labeling and with consistent results. Furthermore, these reagents (made in our lab) have been used and validated extensively in collaborative work across the CEIRS and CIVIC network (manuscripts in preparation). In particular, a successful antibody repertoire sequencing of H1, H3 and H1/H3 HA-probe sorted Bmems has been performed, confirming the HA specificity of the corresponding expressed mAbs.

-I'm confused what the frequencies in 3D and 3E represent

Authors: Fig 3D and 3E show frequency of Vaccine-specific cells in the PB compartment.

-Fig S2 is not evidence of a highly-specific (vs. broadly reactive) vaccine response. Is this a typo?

Authors: We appreciate the reviewer’s attention to detail. This has been corrected.

Figure 6: -Again, a control for the HA staining would be useful - the frequency of HA-binding for the vaccine strain are high

Authors: The frequencies herein reported are in line with existing literature for influenza specific cells in the class-switched compartment after vaccination (ranging from 0.1 to 2% in most participants, but some can show up to 10%) (See DOI: 10.1128/JVI.00169-19, 10.1073/pnas.1414070111).

-"However, when tracking the dynamics of HA-specific CS-Bmems over time, there was an increase in vaccinespecific CS-Bmems at the cost of reactive cells to historical strains 7-9 days after vaccination (Fig. 6C-D)."

Percentages are being shown here, so the increase in vaccine-specific cells should not be described in a way that suggests historical strain-reactive cells are being lost, since absolute numbers might be the same but other pools may have increased.

Authors: We agree with the reviewer’s comment. Text has been changed to more accurately reflect what has been observed (line 221).

-It should be noted (6I) that the B cell compartment in young vs. old individuals is quite different in the proportional representation of lymphocyte subsets, and that this difference could impact ASC differentiation of in vitro stimulated lymphocytes. This is a significant caveat to the experiment.

Authors: We understand the reviewer’s concern, however, we did not observe any impact on total IgG secretion (not antigen specific) between young and elderly participants after in vitro differentiation. Similarly, in a previous sectional study with a much larger sample size, age did not impact total IgG secretion following in vitro differentiation. Therefore, the differences observed seem to be specific to influenza reactive cells. 

-What does "IgG conditioned" mean? This is not made clear from the text.

Authors: It should say “conditioned supernatants”, meaning media collected 7-9 days after in vitro differentiation (See line 232).

Figure 7: -HA staining again looks suspect - over 7% of ABCs binding the vaccine probe – this is rather high (unless there is an aspect of pre-gating not evident from the figure).

Authors: We do not have a reference for the expected frequencies of influenza-specific B-cells in the DN compartment after influenza vaccination, as to the best of our knowledge we are the first to report it. However, since the frequency of rHA-specific cells in the CS-Bmem and PB compartments are aligned with previous literature, we have no reason to question our data. Studies are underway for a detailed transcriptomic characterization of these B-cells, as well as the functional properties of the antibodies they represent.

---

## [Decision Letter · Decision Letter 1]

28 May 2021

PONE-D-20-36767R1

Impaired memory B-cell recall responses in the elderly following recurrent influenza vaccination

PLOS ONE

Dear Dr. Ross,

Thank you for submitting your manuscript to PLOS ONE. After careful consideration, we feel that it has merit but does not fully meet PLOS ONE’s publication criteria as it currently stands. Therefore, we invite you to submit a revised version of the manuscript that addresses the points raised during the review process.

During the revision process, please more clearly detail the sample size issues encountered within the body of the text, as requested by the reviewer.

We look forward to receiving your revised manuscript.

Kind regards,

Victor C Huber

Academic Editor

PLOS ONE

Journal Requirements:

Reviewers' comments:

Reviewer's Responses to Questions

**Comments to the Author**

1. If the authors have adequately addressed your comments raised in a previous round of review and you feel that this manuscript is now acceptable for publication, you may indicate that here to bypass the “Comments to the Author” section, enter your conflict of interest statement in the “Confidential to Editor” section, and submit your "Accept" recommendation.

Reviewer #1: (No Response)

2. Is the manuscript technically sound, and do the data support the conclusions?

Reviewer #1: Yes

3. Has the statistical analysis been performed appropriately and rigorously? 

Reviewer #1: No

4. Have the authors made all data underlying the findings in their manuscript fully available?

Reviewer #1: Yes

5. Is the manuscript presented in an intelligible fashion and written in standard English?

Reviewer #1: Yes

6. Review Comments to the Author

Reviewer #1: PONE-D-20-36767-R1

Impaired memory B-cell recall responses in the elderly following recurrent influenza vaccination

Abreu et al

The authors’ response to reviewers clarifies and justifies the manuscript. In most cases, these changes are included in the document. However, although in their response the authors agree with reviewers that “with only 12 individuals, it is impossible to see statistically significant trends, and individual variation tends to stand-out”, several of their these qualifications don’t seem to have made it into the manuscript itself. The authors continue to make fairly strong claims in the Results and Discussion, including several that they agree lack statistical support (“due to the small sample size we cannot back-up this conclusion with a statistical value”). Presenting these findings as observations is fine in this context, but there should be stronger and more clear qualifiers in the text emphasizing that these are not strongly supported. As it is, a casual reader could easily believe that the observations are much more rigorous than they actually are. Simply moving some of the qualifications from the response to reviewers, into the actual text, should suffice.

7. PLOS authors have the option to publish the peer review history of their article (what does this mean?). If published, this will include your full peer review and any attached files.

Reviewer #1: No

---

## [Author Response · Author response to Decision Letter 1]

24 Jun 2021

June 15, 2021

PLoS One

Editor

Dear Editor: We are resubmitting revised manuscript, # PONE-D-20-36767R1 entitled “Impaired memory B-cell recall responses in the elderly following recurrent influenza vaccination”. to address the referee’s minor comments to our second submission. We hope the paper is now acceptable for publication. 

Best regards,

Ted M. Ross, Ph.D.

GRA Eminent Scholar in Infectious Diseases

Director - Center for Vaccines and Immunology

Professor - Department of Infectious Diseases

University of Georgia

---

## [Editor Report · Decision Letter 2]

28 Jun 2021

Impaired memory B-cell recall responses in the elderly following recurrent influenza vaccination

PONE-D-20-36767R2

Dear Dr. Ross,

We’re pleased to inform you that your manuscript has been judged scientifically suitable for publication and will be formally accepted for publication once it meets all outstanding technical requirements.

Kind regards,

Victor C Huber

Academic Editor

PLOS ONE
---

## [Editor Report · Acceptance letter]

26 Jul 2021

PONE-D-20-36767R2 

Impaired memory B-cell recall responses in the elderly following recurrent influenza vaccination 

Dear Dr. Ross:

I'm pleased to inform you that your manuscript has been deemed suitable for publication in PLOS ONE. Congratulations! Your manuscript is now with our production department. 

Kind regards, 

on behalf of

Dr. Victor C Huber 

Academic Editor

PLOS ONE